# Neural Approximate Mirror Maps for Constrained Diffusion Models

**Berthy T. Feng**
California Institute of Technology
`bfeng@caltech.edu`

**Ricardo Baptista**
California Institute of Technology
`rsb@caltech.edu`

**Katherine L. Bouman**
California Institute of Technology
`klbouman@caltech.edu`

## Abstract

Diffusion models excel at creating visually-convincing images, but they often struggle to meet subtle constraints inherent in the training data. Such constraints could be physics-based (e.g., satisfying a PDE), geometric (e.g., respecting symmetry), or semantic (e.g., including a particular number of objects). When the training data all satisfy a certain constraint, enforcing this constraint on a diffusion model makes it more reliable for generating valid synthetic data and solving constrained inverse problems. However, existing methods for constrained diffusion models are restricted in the constraints they can handle. For instance, recent work proposed to learn mirror diffusion models (MDMs), but analytical mirror maps only exist for convex constraints and can be challenging to derive. We propose *neural approximate mirror maps* (NAMMs) for general, possibly non-convex constraints. Our approach only requires a differentiable distance function from the constraint set. We learn an approximate mirror map that transforms data into an unconstrained space and a corresponding approximate inverse that maps data back to the constraint set. A generative model, such as an MDM, can then be trained in the learned mirror space and its samples restored to the constraint set by the inverse map. We validate our approach on a variety of constraints, showing that compared to an unconstrained diffusion model, a NAMM-based MDM substantially improves constraint satisfaction. We also demonstrate how existing diffusion-based inverse-problem solvers can be easily applied in the learned mirror space to solve constrained inverse problems.

## 1 Introduction

Many data distributions follow a rule that is not visually obvious. For example, videos of fluid flow obey a partial differential equation (PDE), but a human may find it difficult to discern whether a video agrees with the prescribed PDE. We can characterize such distributions as constrained distributions. Theoretically, a generative model trained on a constrained image distribution should satisfy the constraint, but in practice due to learning and sampling errors (Daras et al., 2024), it may generate visually-convincing images that break the rules. Ensuring constraint satisfaction in spite of such errors would make generative models more reliable for applications such as solving inverse problems.

Diffusion models are popular generative models, but existing approaches for incorporating constraints either restrict the type of constraint or do not scale well. Equivariant (Niu et al., 2020), Riemannian (De Bortoli et al., 2022; Huang et al., 2022), reflected (Lou & Ermon, 2023; Fishman et al., 2023a), log-barrier (Fishman et al., 2023a), and mirror (Liu et al., 2024) diffusion models are all restricted to certain types of constraints, such as symmetry groups (Niu et al., 2020), Riemannian manifolds (De Bortoli et al., 2022; Huang et al., 2022), or convex sets (Fishman et al., 2023a; Liu et al., 2024). Generally speaking, these restrictions are in the service of guaranteeing a hard constraint, whereas a soft constraint offers flexibility and may be sufficient in many scenarios. One could, for example, introduce a guidance term to encourage constraint satisfaction during sampling (Graikos

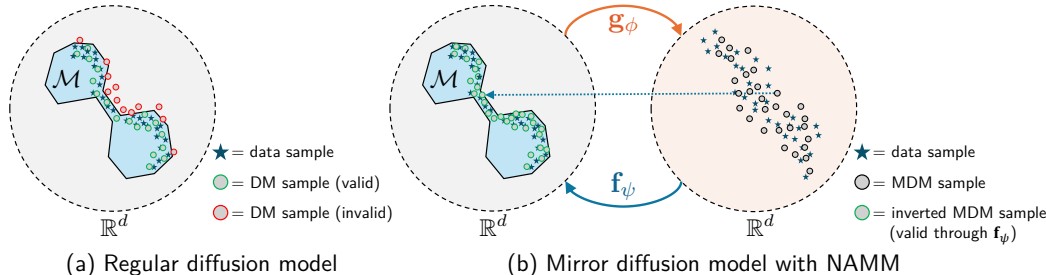

Figure 1: Conceptual illustration. (a) Despite being trained on a data distribution constrained to $\mathcal{M}$, a regular diffusion model (DM) may generate samples that violate the constraint. (b) We propose to learn a neural approximate mirror map (NAMM) that entails a forward map $\mathbf{g}_\phi$ and inverse map $\mathbf{f}_\psi$. The forward map transforms the constrained space into an unconstrained ("mirror") space. Once $\mathbf{g}_\phi$ and $\mathbf{f}_\psi$ are learned, a mirror diffusion model (MDM) can be trained on the pushforward of the data distribution through $\mathbf{g}_\phi$ and its samples mapped back to the constrained space through $\mathbf{f}_\psi$.

et al., 2022; Bansal et al., 2023; Zhang et al., 2023), but so far there does not exist a principled framework to do so. Previous work suggested imposing a soft constraint during training (Daras et al., 2024; Upadhyay et al., 2023) by estimating the clean image at every noisy diffusion step and evaluating its constraint satisfaction. However, approximation (Efron, 2011) of the clean image from the intermediate noisy image is crude at high noise levels and thus unsuitable for constraints that are sensitive to approximation error and noise, such as a PDE constraint. Of course it is possible to penalize invalid generated samples during or after training (Huang et al., 2024), but this approach relies on computationally-expensive simulation steps. Instead, we aim for a flexible approach to impose general constraints by construction.

Our goal is to find an invertible function that maps constrained images into an unconstrained space so that a regular generative model can be trained in the unconstrained space and automatically satisfy the constraint through the inverse function. We propose *neural approximate mirror maps* (NAMMs)[1], which bring the flexibility of soft constraints into the principled framework of mirror diffusion models (Liu et al., 2024). A mirror diffusion model (MDM) allows for training a completely unconstrained diffusion model in a "mirror" space defined by a mirror map. Unconstrained samples from the diffusion model are mapped back to the constrained space via an inverse mirror map. However, invertible mirror maps are challenging or impossible to derive in closed form for general constraints. We address this by jointly optimizing two networks to approximate a mirror map and its inverse.

A NAMM encompasses a (forward) mirror map $\mathbf{g}_\phi$ and its approximate inverse map $\mathbf{f}_\psi$. They are trained so that $\mathbf{f}_\psi \approx \mathbf{g}_\phi^{-1}$, and $\mathbf{f}_\psi$ maps unconstrained points to the constrained space (see Figure 1 for a conceptual illustration). Our method works for any constraint that has a differentiable function to quantify the distance from an image to the constraint set. We parameterize $\mathbf{g}_\phi$ as the gradient of a strongly input-convex neural network (ICNN) (Amos et al., 2017) to satisfy invertibility. We train the NAMM with a cycle-consistency loss (Zhu et al., 2017) to ensure $\mathbf{f}_\psi \approx \mathbf{g}_\phi^{-1}$ and train the inverse map with a constraint loss to ensure $\mathbf{f}_\psi(\tilde{\mathbf{x}})$ is close to the constraint set for all $\tilde{\mathbf{x}}$ that we are interested in (we define this formally in Section 3.1). An MDM can be trained on the pushforward of the data distribution through $\mathbf{g}_\phi$, and its generated samples can be mapped to the constraint set via $\mathbf{f}_\psi$. Although not inherently restricted to diffusion models, our approach maintains the many advantages of diffusion models, including expressive generation, simulation-free training (Song et al., 2021b), and tractable computation of probability densities (Liu et al., 2024). One can also adapt existing diffusion-based inverse solvers for the mirror space and enforce constraints with the inverse map.

Our experiments show improved constraint satisfaction for various physics-based, geometric, and semantic constraints. We also discuss ablation studies and adapt a popular diffusion-based inverse solver to solve constrained inverse problems, in particular data assimilation with PDE constraints.

---

[1]Code can be found at `https://github.com/berthyf96/namm`.

## 2 BACKGROUND

### 2.1 CONSTRAINED GENERATIVE MODELS

Explicitly incorporating a known constraint into a generative model poses benefits such as data efficiency (Ganchev et al., 2010; Batzner et al., 2022), generalization capabilities (Köhler et al., 2020), and feasibility of samples (Giannone et al., 2023). Some methods leverage equivariant neural networks (Satorras et al., 2021; Geiger & Smidt, 2022; Thomas et al., 2018) for symmetry (Allingham et al., 2022; Niu et al., 2020; Klein et al., 2024; Song et al., 2024; Boyda et al., 2021; Rezende et al., 2019; Garcia Satorras et al., 2021; Köhler et al., 2020; Midgley et al., 2024; Dey et al., 2020; Xu et al., 2022; Hoogeboom et al., 2022; Yim et al., 2023) but do not generalize to other types of constraints or generative models (Klein et al., 2024; Song et al., 2024; Boyda et al., 2021; Rezende et al., 2019; Garcia Satorras et al., 2021; Köhler et al., 2020; Midgley et al., 2024; Niu et al., 2020; Xu et al., 2022; Hoogeboom et al., 2022; Yim et al., 2023; Dey et al., 2020; Allingham et al., 2022). Previous methods for constrained diffusion models (De Bortoli et al., 2022; Huang et al., 2022; Fishman et al., 2023a; Lou & Ermon, 2023; Liu et al., 2024) make strong assumptions about the constraint, such as being characterized as a Riemannian manifold (De Bortoli et al., 2022; Huang et al., 2022), having a well-defined reflection operator (Lou & Ermon, 2023) or projection operator (Christopher et al., 2024), or corresponding to a convex constraint set (Fishman et al., 2023a; Liu et al., 2024). Fishman et al. (2023b) proposed a diffusion model that incorporates Metropolis-Hastings steps to work with general constraints, but impractically high rejection rates may occur with constraints that are challenging to satisfy, such as a PDE constraint.

An alternative approach is to introduce a soft constraint penalty when training the generative model (Ganchev et al., 2010; Mann & McCallum, 2007; Chang et al., 2007; Giannone et al., 2023; Daras et al., 2024; Upadhyay et al., 2023). However, evaluating the constraint loss of generated samples during training may be prohibitively expensive. Instead, one could add constraint-violating training examples (Giannone et al., 2023), but it is often difficult to procure useful negative examples. In contrast, our approach does not alter the training objective of the generative model.

### 2.2 MIRROR MAPS

For any convex constraint set $\mathcal{C} \subseteq \mathbb{R}^d$, one can define a *mirror map* that maps from $\mathcal{C}$ to $\mathbb{R}^d$. This is done by defining a *mirror potential* $\phi : \mathcal{C} \to \mathbb{R}$ that is continuously-differentiable and strongly-convex (Bubeck et al., 2015; Tan et al., 2023). The mirror map is the gradient $\nabla\phi : \mathcal{C} \to \mathbb{R}^d$ (Liu et al., 2024). Every mirror map has an inverse $(\nabla\phi)^{-1} : \mathbb{R}^d \to \mathcal{C}$, which, unlike the forward mirror map, is not necessarily the gradient of a strongly-convex function (Zhou, 2018; Tan et al., 2023). Mirror maps have been used for constrained optimization (Beck & Teboulle, 2003) and sampling (Hsieh et al., 2018; Li et al., 2022; Liu et al., 2024). Although true mirror maps exist only for convex constraints, we seek to generalize the concept to learn approximate mirror maps to handle non-convex constraints. Recent work suggested learned mirror maps for convex optimization (Tan et al., 2023) and reinforcement learning (Alfano et al., 2024) but did not tackle constrained generative modeling. Our work proposes a novel training objective for learning mirror maps with the goal of constraining generative models to satisfy arbitrary constraints.

### 2.3 DIFFUSION MODELS

A diffusion model learns to generate new data samples through a gradual denoising process (Sohl-Dickstein et al., 2015; Ho et al., 2020; Song & Ermon, 2019; Song et al., 2021b; Kingma et al., 2021). The diffusion, or noising, process can be modeled as a stochastic differential equation (SDE) (Song et al., 2021b) that induces a time-dependent distribution $p_t$, where $p_0 = p_{\text{data}}$ (the target data distribution) and $p_T \approx \mathcal{N}(\mathbf{0}, \mathbf{I})$. The diffusion model learns to reverse the denoising process by modeling the *score function* of $p_t$, defined as $\nabla_{\mathbf{x}} \log p_t(\mathbf{x})$. In the context of imaging problems, the score function is often parameterized using a convolutional neural network (CNN) with parameters $\theta$, which we denote by $\mathbf{s}_\theta$. The score model is trained with a denoising-based objective (Hyvärinen & Dayan, 2005; Vincent, 2011; Song et al., 2021b) that allows for simulation-free training (i.e., simulating the forward noising process is not necessary during training). The score function appears in a reverse-time SDE (Song et al., 2021b; Anderson, 1982) that can be used to sample from the clean image distribution $p_0$ by first sampling a noise image from $\mathcal{N}(\mathbf{0}, \mathbf{I})$ and then gradually denoising it. Our work addresses the problem that diffusion models are often not sensitive to visually-subtle

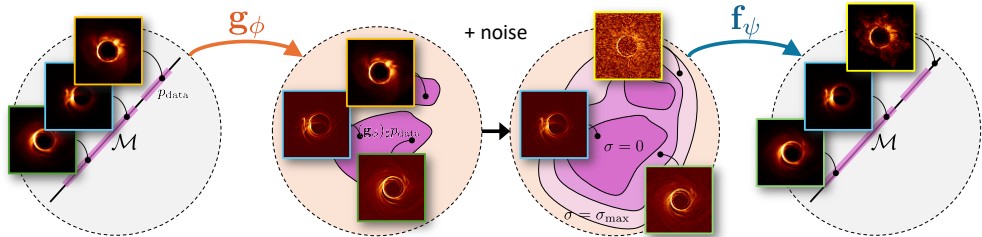

Figure 2: NAMM training illustration. Given data that lie on a constraint manifold $\mathcal{M}$ (e.g., the hyperplane of images with the same total brightness), we jointly train an approximate mirror map $\mathbf{g}_\phi$ and its approximate inverse $\mathbf{f}_\psi$. After mapping data $\mathbf{x} \sim p_{\text{data}}$ to the mirror space as $\mathbf{g}_\phi(\mathbf{x})$, we perturb them with additive Gaussian noise whose standard deviation can be anywhere between $0$ and $\sigma_{\text{max}}$. The inverse map $\mathbf{f}_\psi$ is trained to map these perturbed samples back onto $\mathcal{M}$.

constraints on the data. Our proposed NAMM allows for a diffusion model to be trained in an unconstrained space yet satisfy the desired constraint by construction.

## 3 METHOD

We now describe neural approximate mirror maps (NAMMs). We focus on diffusion models, but any generative model can be trained in the learned mirror space (see Appendix A for results with a VAE). We denote the constrained image distribution by $p_{\text{data}}$ and the (not necessarily convex) constraint set by $\mathcal{M} \subseteq \mathbb{R}^d$. Images in the constrained and mirror spaces are denoted by $\mathbf{x}$ and $\tilde{\mathbf{x}}$, respectively. The pushforward of the data distribution $p_{\text{data}}$ through a mirror map $\mathbf{g}_\phi$ is denoted by $(\mathbf{g}_\phi)_\sharp p_{\text{data}}$.

### 3.1 LEARNING THE FORWARD AND INVERSE MIRROR MAPS

Let $\mathbf{g}_\phi$ and $\mathbf{f}_\psi$ be the neural networks modeling the forward and inverse mirror maps, respectively, where $\phi$ and $\psi$ are their parameters. We formulate the following learning problem:

$$\phi^*, \psi^* \in \underset{\phi, \psi}{\arg\min} \left\{ \mathcal{L}_{\text{cycle}}(\mathbf{g}_\phi, \mathbf{f}_\psi) + \lambda_{\text{constr}} \mathcal{L}_{\text{constr}}(\mathbf{g}_\phi, \mathbf{f}_\psi) + \lambda_{\text{reg}} \mathcal{R}(\mathbf{g}_\phi) \right\}, \tag{1}$$

where $\mathcal{L}_{\text{cycle}}$ encourages $\mathbf{g}_\phi$ and $\mathbf{f}_\psi$ to be inverses of each other; $\mathcal{L}_{\text{constr}}$ encourages $\mathbf{f}_\psi$ to map unconstrained points back to the constraint set; and $\mathcal{R}$ is a regularization term to ensure there is a unique solution for the maps. Here $\lambda_{\text{constr}}, \lambda_{\text{reg}} \in \mathbb{R}_{>0}$ are scalar hyperparameters.

A true inverse mirror map satisfies cycle consistency and constraint satisfaction on all of $\mathbb{R}^d$, so ideally $\mathbf{f}_\psi(\tilde{\mathbf{x}}) = \mathbf{g}_\phi^{-1}(\tilde{\mathbf{x}})$ and $\mathbf{f}_\psi(\tilde{\mathbf{x}}) \in \mathcal{M}$ for all $\tilde{\mathbf{x}} \in \mathbb{R}^d$. But since it would be computationally infeasible to optimize $\mathbf{f}_\psi$ over all possible points in $\mathbb{R}^d$, we instead optimize it over distributions that we would expect the inverse map to face in practice in the context of generative models. That is, we only need $\mathbf{f}_\psi$ to be valid for samples from an MDM trained on $(\mathbf{g}_\phi)_\sharp p_{\text{data}}$, which we refer to as the *mirror distribution*. To make $\mathbf{f}_\psi$ robust to learning/sampling error of the MDM, we consider a sequence of noisy distributions in the mirror space, each corresponding to adding Gaussian noise to samples from $(\mathbf{g}_\phi)_\sharp p_{\text{data}}$, which, for a maximum perturbation level $\sigma_{\text{max}}$, we denote by

$$\left( (\mathbf{g}_\phi)_\sharp p_{\text{data}} * \mathcal{N}(\mathbf{0}, \sigma^2 \mathbf{I}) \right)_{\sigma \in [0, \sigma_{\text{max}}]}. \tag{2}$$

We train $\mathbf{f}_\psi$ to be a valid inverse mirror map only for points from these noisy mirror distributions. Since we do not know *a priori* how much noise the MDM samples will contain, we consider all possible noise levels up to $\sigma_{\text{max}}$ for robustness (see Appendix D.2 for an ablation study of this choice).

We define a **cycle-consistency loss** (Zhu et al., 2017) that covers the forward and inverse directions and evaluates the inverse direction for the entire sequence of distributions defined in Equation 2:

$$\mathcal{L}_{\text{cycle}}(\mathbf{g}_\phi, \mathbf{f}_\psi) := \mathbb{E}_{\mathbf{x} \sim p_{\text{data}}} \Bigg[ \left\| \mathbf{x} - \mathbf{f}_\psi\left(\mathbf{g}_\phi(\mathbf{x})\right) \right\|_1$$
$$+ \int_0^{\sigma_{\text{max}}} \mathbb{E}_{\mathbf{z} \sim \mathcal{N}(\mathbf{0}, \mathbf{I})} \left[ \left\| \mathbf{g}_\phi(\mathbf{x}) + \sigma \mathbf{z} - \mathbf{g}_\phi\left(\mathbf{f}_\psi(\mathbf{g}_\phi(\mathbf{x}) + \sigma \mathbf{z})\right) \right\|_1 \right] \Bigg]. \tag{3}$$

Let $\ell_{\text{constr}} : \mathbb{R}^d \to \mathbb{R}_{\geq 0}$ be a differentiable *constraint distance* that measures the distance from an input image to the constraint set. We define the following **constraint loss** to encourage $\mathbf{f}_\psi$ to map points from the noisy mirror distributions (Equation 2) to the constraint set:

$$\mathcal{L}_{\text{constr}}(\mathbf{g}_\phi, \mathbf{f}_\psi) := \mathbb{E}_{\mathbf{x} \sim p_{\text{data}}} \left[ \int_0^{\sigma_{\max}} \mathbb{E}_{\mathbf{z} \sim \mathcal{N}(\mathbf{0}, \mathbf{I})} \left[ \ell_{\text{constr}} \left( \mathbf{f}_\psi (\mathbf{g}_\phi(\mathbf{x}) + \sigma \mathbf{z}) \right) \right] \mathrm{d}\sigma \right]. \tag{4}$$

To ensure a unique solution, we **regularize** $\mathbf{g}_\phi$ to be close to the identity function:

$$\mathcal{R}(\mathbf{g}_\phi) := \mathbb{E}_{\mathbf{x} \sim p_{\text{data}}} \left[ \| \mathbf{x} - \mathbf{g}_\phi(\mathbf{x}) \|_1 \right]. \tag{5}$$

We use Monte-Carlo to approximate the expectations in the objective over the noisy mirror distributions with $\sigma \sim \mathcal{U}([0, \sigma_{\max}])$ and approximately solve Equation 1 with stochastic gradient descent.

**Architecture** We parameterize $\mathbf{g}_\phi$ as the gradient of an input-convex neural network (ICNN) following the implementation of Tan et al. (2023). For convex constraints, this satisfies the theoretical requirement that $\mathbf{g}_\phi$ be the gradient of a strongly-convex function. Even for non-convex constraints, this choice brings practical benefits, as we discuss in Section 4.3. We note that $\mathbf{g}_\phi$ is not a true mirror map since $\mathcal{M}$ is not assumed to be convex, and $\mathbf{g}_\phi$ is defined on all of $\mathbb{R}^d$ instead of just on $\mathcal{M}$. We parameterize $\mathbf{f}_\psi$ as a ResNet-based CNN similar to the one used in CycleGAN (Zhu et al., 2017).

## 3.2 LEARNING THE MIRROR DIFFUSION MODEL

Similarly to Liu et al. (2024), we train an MDM on the mirror distribution $(\mathbf{g}_\phi)_\sharp p_{\text{data}}$ and map its samples to the constrained space through $\mathbf{f}_\psi$. In particular, we train a score model $\mathbf{s}_\theta$ with the following denoising score matching objective in the learned mirror space (defined as the range of $\mathbf{g}_\phi$):

$$\theta^* \in \arg\min_\theta \mathbb{E}_t \left\{ \lambda(t) \mathbb{E}_{\tilde{\mathbf{x}}(0) \sim (\mathbf{g}_\phi)_\sharp p_{\text{data}}} \mathbb{E}_{\tilde{\mathbf{x}}(t) | \tilde{\mathbf{x}}(0)} \left[ \left\| \tilde{\mathbf{s}}_\theta (\tilde{\mathbf{x}}(t), t) - \nabla_{\tilde{\mathbf{x}}(t)} \log p_{0t} (\tilde{\mathbf{x}}(t) \mid \tilde{\mathbf{x}}(0)) \right\|_2^2 \right] \right\}, \tag{6}$$

where $\tilde{\mathbf{x}}(0) \sim (\mathbf{g}_\phi)_\sharp p_{\text{data}}$ is obtained as $\tilde{\mathbf{x}}(0) := \mathbf{g}_\phi(\mathbf{x}(0))$ for $\mathbf{x}(0) \sim p_{\text{data}}$. Here $p_{0t}$ denotes the transition kernel from $\tilde{\mathbf{x}}(0)$ to $\tilde{\mathbf{x}}(t)$ under the diffusion SDE, and $\lambda(t) \in \mathbb{R}_{>0}$ is a time-dependent weight. To sample new images, we sample $\tilde{\mathbf{x}}(T) \sim \mathcal{N}(\mathbf{0}, \mathbf{I})$, run reverse diffusion in the mirror space, and map the resulting $\tilde{\mathbf{x}}(0)$ to the constrained space via $\mathbf{f}_\psi$.

## 3.3 FINETUNING THE INVERSE MIRROR MAP

The inverse map $\mathbf{f}_\psi$ is trained with samples from the noisy mirror distributions in Equation 2, but we ultimately wish to evaluate $\mathbf{f}_\psi$ with samples from the MDM. To reduce the distribution shift, it may be helpful to finetune $\mathbf{f}_\psi$ with MDM samples. We generate a training dataset of samples $\tilde{\mathbf{x}}$ from the MDM and then finetune the inverse map to deal with such samples specifically. In the following finetuning objective, we replace $\tilde{\mathbf{x}} \sim (\mathbf{g}_\phi)_\sharp p_{\text{data}}$ with $\tilde{\mathbf{x}} \sim p_\theta$, where $p_\theta$ is the distribution of MDM samples in the mirror space:

$$\psi^* = \arg\min_\psi \left\{ \mathbb{E}_{\mathbf{x} \sim p_{\text{data}}} \| \mathbf{x} - \mathbf{f}_\psi(\mathbf{g}_\phi(\mathbf{x})) \|_1 \right.$$
$$\left. + \mathbb{E}_{\tilde{\mathbf{x}} \sim p_\theta} \left[ \int_0^{\sigma_{\max}} \mathbb{E}_{\mathbf{z} \sim \mathcal{N}(\mathbf{0}, \mathbf{I})} \left[ \| \tilde{\mathbf{x}} + \sigma \mathbf{z} - \mathbf{g}_\phi(\mathbf{f}_\psi(\tilde{\mathbf{x}} + \sigma \mathbf{z})) \|_1 + \lambda_{\text{constr}} \ell_{\text{constr}} \left( \mathbf{f}_\psi(\tilde{\mathbf{x}} + \sigma \mathbf{z}) \right) \right] \mathrm{d}\sigma \right] \right\}. \tag{7}$$

Finetuning essentially tailors $\mathbf{f}_\psi$ to the MDM. The original objective assumes that the MDM will sample Gaussian-perturbed images from the mirror distribution, but in reality it samples from a slightly different distribution. As the ablation study in Section 4.3 shows, finetuning is not an essential component of the method; we suggest it as an optional step for when it is critical to optimize the constraint distance metric.

## 4 RESULTS

We present experiments with constraints ranging from physics-based to semantic. For the considered examples, our method achieves from $38\%$ to as much as $96\%$ improvement in constraint satisfaction upon a vanilla DM trained on the same data (see Table 1). Appendices D.1 and F provide implementation and constraint details, respectively. The following paragraphs introduce the demonstrated constraints $\ell$. For each we consider an image dataset for which the constraint is physically meaningful.

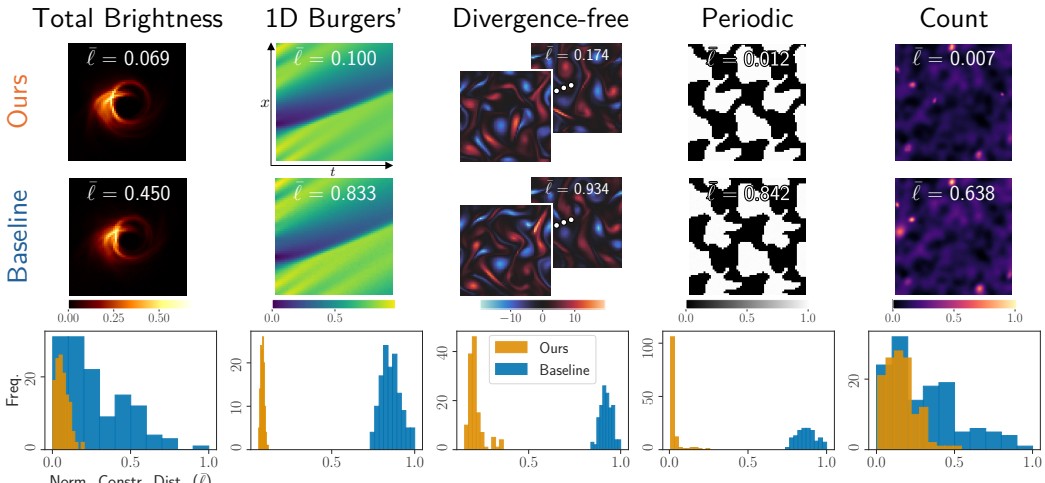

Figure 3: Improved constraint satisfaction. Samples from our approach are nearly indistinguishable from baseline samples, yet there is a significant difference in their distances from the constraint set. The baseline is a DM trained on the original constrained dataset. Our approach is to train a NAMM and then an MDM in the mirror space induced by $\mathbf{g}_\phi$. Samples are obtained by sampling from the MDM and then passing samples through $\mathbf{f}_\psi$. The histograms show normalized constraint distances $\bar{\ell}$ of 128 samples (normalized so that each constraint has a maximum of 1 across the samples from both methods). Our results are from the finetuned NAMM. For each constraint, we made sure that the DM was trained for at least as long as the NAMM, MDM, and finetuned NAMM combined.

**Total brightness**   In astronomical imaging, even if a source's structure is unknown *a priori*, its total brightness, or total flux, is often well constrained (EHTC, 2019). We define $\ell_{\text{flux}}(\mathbf{x})$ as the absolute difference between $\sum_{i=1}^d \mathbf{x}_i$ and the true total brightness. We demonstrate with a dataset of $64 \times 64$ images of black-hole simulations (Wong et al., 2022) whose pixel values sum to 120. While this constraint is a simple warmup example, generic diffusion models perform surprisingly poorly on it.

**1D Burgers'**   We consider Burgers' equation (Bateman, 1915; Burgers, 1948) for a 1D viscous fluid, representing the discretized solution as an $n_x \times n_t$ image $\mathbf{x}$, where $n_x$ and $n_t$ are the numbers of grid points in space and time, respectively. The distance $\ell_{\text{burgers}}(\mathbf{x})$ compares each 1D state in the image to the PDE solver's output given the previous state (based on Crank-Nicolson time-discretization (Crank & Nicolson, 1947; Kidger, 2022)). The dataset consists of $64 \times 64$ images of Crank-Nicolson solutions with Gaussian random fields as initial conditions.

**Divergence-free**   A time-dependent 2D velocity field $\mathbf{u} = \mathbf{u}(x, y, t)$ is called *divergence-free* or *incompressible* if $\nabla \cdot \mathbf{u} = 0$. We define the constraint distance $\ell_{\text{div}}$ as the $\ell^1$-norm of the divergence and demonstrate this constraint with 2D Kolmogorov flows (Chandler & Kerswell, 2013; Boffetta & Ecke, 2012; Rozet & Louppe, 2024). We represent the trajectory of the 2D velocity, discretized in space-time, as a two-channel (for both velocity components) image $\mathbf{x}$ with the states appended sequentially. We used jax-cfd (Kochkov et al., 2021) to generate trajectories of eight $64 \times 64$ states and appended them in a $2 \times 4$ pattern to create $128 \times 256$ images.

**Periodic**   We consider images $\mathbf{x}$ that are periodic tilings of a unit cell. This type of symmetry appears in materials science, such as when constructing metamaterials out of unit cells (Ogren et al., 2024). We use a distance function $\ell_{\text{periodic}}$ that compares all pairs of tiles in the image and computes the average $\ell^1$-norm of their differences. We created a dataset of $64 \times 64$ images (composed of $32 \times 32$ unit cells tiled in a $2 \times 2$ fashion) using code from Ogren et al. (2024).

**Count**   Generative models can sometimes generate an incorrect number of objects (Paiss et al., 2023). We formulate a differentiable count constraint by relying on a CNN to estimate the count of a particular object in an image $\mathbf{x}$. Note that using a neural network leads to a non-analytical and highly non-convex constraint. Letting $f_{\text{CNN}} : \mathbb{R}^d \to \mathbb{R}$ be the trained counting CNN, we use the distance function $\ell_{\text{count}}(\mathbf{x}) := |f_{\text{CNN}}(\mathbf{x}) - \bar{c}|$ for a target count $\bar{c}$. The dataset consists of $128 \times 128$ simulated images of exactly eight (8) radio galaxies with background noise (Connor et al., 2022).

Figure 4: Training efficiency. For each method, we clocked the total compute time during training (ignoring validation and I/O operations) and here plot the mean $\pm$ std. dev. of the constraint distances $\ell$ of 128 generated samples at each checkpoint. The MDM training curve ("Ours w/o FT") is offset by the time it took to train the NAMM. The finetuning curve ("Ours") is offset by the time it took to train the NAMM and MDM and generate finetuning data. For most constraints, the DM has consistently higher constraint distance without any sign of converging to the same performance as that of the MDM. For the count constraint, the MDM performs on par with the DM, but finetuning noticeably accelerates constraint satisfaction. Each run was done on the same hardware ($4\times$ A100 GPUs).

## 4.1 IMPROVED CONSTRAINT SATISFACTION AND TRAINING EFFICIENCY

First and foremost, we verify that our approach leads to better constraint satisfaction than a vanilla diffusion model (DM). We evaluate constraint satisfaction by computing the average constraint distance of generated samples. Since the constraint distance is non-negative, an average constraint distance of 0 implies that the constraint is satisfied almost surely.

For each constraint, we trained a NAMM on the corresponding dataset and then trained an MDM on the pushforward of the dataset through the learned $\mathbf{g}_\phi$. We show results from a finetuned NAMM, but as shown in Section 4.3, finetuning is often not necessary. The baseline DM was trained on the original dataset. Figure 3 highlights that MDM samples inverted through $\mathbf{f}_\psi$ are much closer to the constraint set than DM samples despite being visually indistinguishable. For the total brightness, 1D Burgers', divergence-free, and periodic constraints, there is a significant gap between our distribution of constraint distances and the baseline's. The gap is smaller for the count constraint, which may be due to difficulties in identifying and learning the mirror map for a highly non-convex constraint.

Furthermore, our approach achieves better constraint satisfaction in less training time. In Figure 4, we plot constraint satisfaction as a function of compute time, comparing our approach (with and without finetuning) to the DM. Accounting for the time it takes to train the NAMM, our MDM achieves much lower constraint distances than the DM for the three physics-based constraints and the periodic constraint, often reaching a level that the DM struggles to achieve. For the count constraint, we find that finetuning is essential for improving constraint satisfaction, and it is more time-efficient to finetune the inverse map than to continue training the MDM.

## 4.2 SOLVING CONSTRAINED INVERSE PROBLEMS WITH MIRROR DPS

Many methods have been proposed to use a pretrained diffusion model to sample images from the posterior distribution $p(\mathbf{x} \mid \mathbf{y}) \propto p(\mathbf{y} \mid \mathbf{x})p(\mathbf{x})$ (Choi et al., 2021; Graikos et al., 2022; Song et al., 2022; Jalal et al., 2021; Kawar et al., 2022; Song et al., 2023), given measurements $\mathbf{y} \in \mathbb{R}^m$ and a diffusion-model prior $p(\mathbf{x})$. One of the most popular methods is diffusion posterior sampling (DPS) (Chung et al., 2022). To adapt DPS for the mirror space, we simply evaluate the measurement likelihood on inverted mirror images $\mathbf{f}_\psi(\tilde{\mathbf{x}})$ instead of on images $\mathbf{x}$ in the original space.

We demonstrate mirror DPS on data assimilation, an inverse problem that aims to recover the hidden state of a dynamical system given imperfect observations of the state. In Figure 5, we show results for data assimilation of a 1D Burgers' system and a divergence-free Kolmogorov flow given a few noisy state observations, which can be essentially formulated as a denoise-and-inpaint problem. For each case, we used mirror DPS with the corresponding NAMM-based MDM. We include two baselines: (1) vanilla DPS with the DM and (2) constraint-guided DPS (CG-DPS) with the DM. The latter incorporates the constraint distance as an additional likelihood term. As Figure 5 shows, our approach leads to notably less constraint violation (i.e., less deviation from the PDE or less divergence) than both baselines. Appendix D.3 shows that our method consistently outperforms the baselines for different measurement-likelihood and constraint-guidance weights used in DPS and CG-DPS.

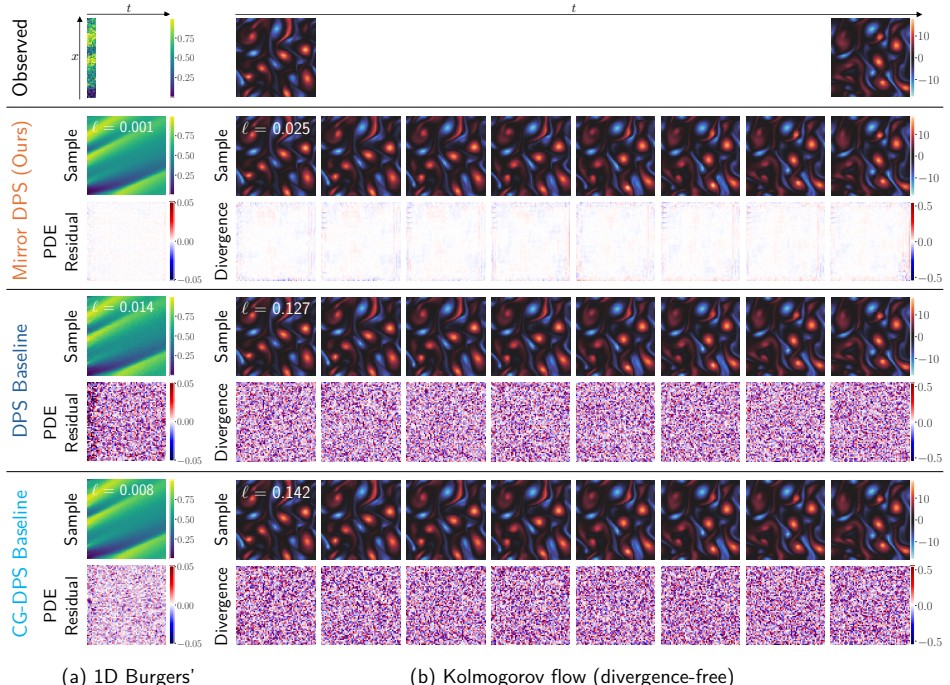

(a) 1D Burgers'          (b) Kolmogorov flow (divergence-free)

Figure 5: Data assimilation. We used the same finetuned NAMM, MDM, and DM checkpoints as in Fig. 3. (a) Given noisy observations of the first eight states, we sampled possible full trajectories of a 1D Burgers' system. Our solutions have smaller deviation from the PDE than samples obtained with DPS, even those of constraint-guided DPS (CG-DPS). (b) The task is to infer the full Kolmogorov flow from noisy observations of the first and last states. Our solution has significantly less divergence.

## 4.3 ABLATION STUDIES

**Finetuning** Table 1 shows the improvement in constraint satisfaction after finetuning $\mathbf{f}_\psi$ while also verifying that the generated distribution stays close to the true data distribution. We use maximum mean discrepancy (MMD) (Gretton et al., 2012) and Kernel Inception Distance (KID) (Bińkowski et al., 2018) as measures of distance between distributions. MMD evaluates distance in a feature space defined by a Gaussian kernel, and KID uses Inception v3 (Szegedy et al., 2015) features (see Appendix E for details). Finetuning does not notably change the distribution-matching accuracy of the MDM and in some cases improves it while improving constraint satisfaction. Compared to a vanilla DM, our approach before and after finetuning does not lead to significantly different MMD and even gives better KID while significantly improving constraint distance.

| | | Total Brightness | 1D Burgers' | Divergence-free | Periodic | Count |
|---|---|---|---|---|---|---|
| Ours w/o FT | CD ($\downarrow$) | $0.57 \pm 0.48$ | $0.09 \pm 0.01$ | $2.51 \pm 0.21$ | $0.04 \pm 0.05$ | $0.71 \pm 0.53$ |
| | MMD ($\downarrow$) | $0.0957 \pm 0.0130$ | $0.1225 \pm 0.0096$ | $0.0664 \pm 0.0027$ | $0.0760 \pm 0.0058$ | $0.1716 \pm 0.0062$ |
| | KID ($\downarrow$) | $\mathbf{0.0026} \pm 0.0007$ | $0.0040 \pm 0.0004$ | $\mathbf{0.0035} \pm 0.0004$ | $0.0018 \pm 0.0005$ | $\mathbf{0.0367} \pm 0.0010$ |
| Ours | CD ($\downarrow$) | $\mathbf{0.49} \pm 0.35$ | $\mathbf{0.04} \pm 0.01$ | $\mathbf{1.57} \pm 0.31$ | $\mathbf{0.04} \pm 0.06$ | $\mathbf{0.35} \pm 0.27$ |
| | MMD ($\downarrow$) | $0.1023 \pm 0.0131$ | $0.1291 \pm 0.0096$ | $0.0781 \pm 0.0023$ | $0.0758 \pm 0.0058$ | $0.1978 \pm 0.0062$ |
| | KID ($\downarrow$) | $0.0027 \pm 0.0008$ | $\mathbf{0.0022} \pm 0.0004$ | $0.0058 \pm 0.0006$ | $\mathbf{0.0014} \pm 0.0005$ | $0.0570 \pm 0.0014$ |
| Baseline | CD ($\downarrow$) | $2.20 \pm 1.61$ | $0.45 \pm 0.03$ | $6.96 \pm 0.27$ | $1.04 \pm 0.08$ | $0.56 \pm 0.44$ |
| | MMD ($\downarrow$) | $\mathbf{0.0956} \pm 0.0099$ | $\mathbf{0.0621} \pm 0.0091$ | $\mathbf{0.0595} \pm 0.0026$ | $\mathbf{0.0533} \pm 0.0043$ | $\mathbf{0.1276} \pm 0.0066$ |
| | KID ($\downarrow$) | $0.0462 \pm 0.0016$ | $0.2308 \pm 0.0028$ | $0.0053 \pm 0.0007$ | $0.0064 \pm 0.0004$ | $0.1084 \pm 0.0015$ |

Table 1: Effect of finetuning. Constr. dist. (CD) = $100\lambda_{\text{constr}}\ell$. The improvements in mean CD are (left to right, comparing "Ours" to "Baseline"): 78%, 91%, 77%, 96%, 38% for five problems. For all metrics, mean $\pm$ std. dev. is estimated with 10000 samples. In terms of MMD/KID, finetuning does not significantly impact distribution-matching accuracy but improves constraint distance. DM baseline results are shown for comparison. According to MMD, the baseline gives better distribution-matching accuracy; according to KID, our approach captures the true data distribution better.

**Constraint loss** There are two hyperparameters for the constraint loss in Equation 1: $\sigma_{\max}$ determines how much noise to add to samples from the mirror distribution, and $\lambda_{\mathrm{constr}}$ is the weight of the constraint loss. Intuitively, a higher $\sigma_{\max}$ means that the inverse map $\mathbf{f}_\psi$ must map larger regions of $\mathbb{R}^d$ back to the constraint set, making its learning objective more challenging. We would expect $\mathbf{g}_\phi$ to cooperate by maintaining a reasonable SNR in the noisy mirror distributions. Figure 6 shows how increasing $\sigma_{\max}$ causes $\mathbf{g}_\phi(\mathbf{x})$ for $\mathbf{x} \sim p_{\mathrm{data}}$ to have larger magnitudes so that the added noise will not hide the signal. However, setting $\sigma_{\max}$ too high can worsen constraint satisfaction, perhaps due to the challenge of mapping a larger region of $\mathbb{R}^d$ back to the constraint set. On the flip side, setting $\sigma_{\max}$ too low can worsen constraint satisfaction because of poor robustness of $\mathbf{f}_\psi$. Meanwhile, increasing $\lambda_{\mathrm{constr}}$ for the same $\sigma_{\max}$ leads to lower constraint distance, although there is a tradeoff between constraint distance and cycle-consistency inherent in the NAMM objective (Equation 1).

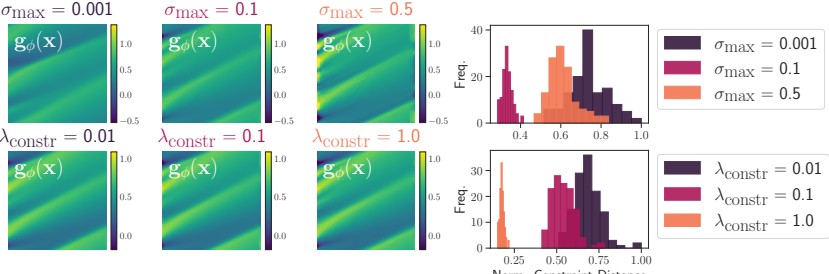

Figure 6: Effect of $\sigma_{\max}$ and $\lambda_{\mathrm{constr}}$, demonstrated with 1D Burgers'. **First row:** as $\sigma_{\max}$ increases (keeping $\lambda_{\mathrm{constr}} = 1.0$), the mirror image $\mathbf{g}_\phi(\mathbf{x})$ for $\mathbf{x} \sim p_{\mathrm{data}}$ increases in magnitude to maintain a similar SNR. Histograms show constraint distances of 128 inverted MDM samples, normalized to have a maximum of 1 across samples from all three settings. Decreasing $\sigma_{\max}$ from 0.5 to 0.1 improves the constraint distances, but further lowering $\sigma_{\max}$ to 0.001 causes them to go back up. This indicates a tradeoff between robustness and performance of $\mathbf{f}_\psi$. **Second row:** as $\lambda_{\mathrm{constr}}$ increases (keeping $\sigma_{\max} = 0.1$), $\mathbf{g}_\phi(\mathbf{x})$ does not change as much as when increasing $\sigma_{\max}$, but the constraint distances decrease (with a tradeoff in cycle consistency). For all three settings, the same number of NAMM and MDM epochs was used as in Fig. 3 but without finetuning.

**Mirror map parameterization** Figure 7 compares parameterizing $\mathbf{g}_\phi$ as the gradient of an ICNN versus as a ResNet-based CNN. We demonstrate how the mirror space changes when parameterizing the forward map as a ResNet-based CNN. The mirror space becomes less regularized, leading to worse constraint satisfaction of the MDM, perhaps because the MDM struggles to learn a less-regularized mirror space. Thus, even when the constraint is non-convex and there are no theoretical reasons to use an ICNN, it may still be practically favorable.

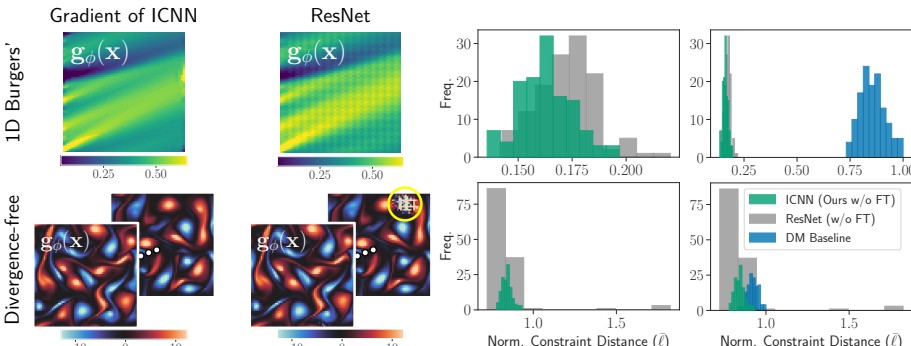

Figure 7: Architecture of $\mathbf{g}_\phi$: gradient of ICNN vs. ResNet-based CNN. Both approaches preserve visual structure in the mirror space, but the ResNet causes irregularities, such as the patch circled in yellow. The histograms show the normalized constraint distances $\bar{\ell}$ of 128 inverted MDM samples (without finetuning). DM histograms from Fig. 3 are shown for comparison. An ICNN leads to better constraint satisfaction with fewer outliers. We trained the NAMM and MDM for the same number of epochs as in Fig. 3 without finetuning. We found that even with finetuning, a ResNet-based forward map leads to worse constraint satisfaction or noticeably worse visual quality of generated samples.

## 5  CONCLUSION

We have proposed a method for constrained diffusion models that minimally restricts the generative model and constraint. A NAMM consists of a mirror map $\mathbf{g}_\phi$ and its approximate inverse $\mathbf{f}_\psi$, which is robust to noise added to samples in the mirror space induced by $\mathbf{g}_\phi$. One can train a mirror diffusion model in this mirror space to generate samples that are constrained by construction via the inverse map. We have validated that our method provides significantly better constraint satisfaction than a vanilla diffusion model on physics-based, geometric, and semantic constraints and have shown its utility for solving constrained inverse problems. Our work establishes that NAMMs effectively rein in generative models according to visually-subtle yet physically-meaningful constraints.

## ACKNOWLEDGMENTS

The authors would like to thank Rob Webber, Jamie Smith, Dmitrii Kochkov, Alex Ogren, and Florian Schäfer for their helpful discussions. BTF and KLB acknowledge support from the NSF GRFP, NSF Award 2048237, NSF Award 2034306, the Pritzker Award, and the Amazon AI4Science Partnership Discovery Grant. RB acknowledges support from the US Department of Energy AEOLUS center (award DE-SC0019303), the Air Force Office of Scientific Research MURI on "Machine Learning and Physics-Based Modeling and Simulation" (award FA9550-20-1-0358), and a Department of Defense (DoD) Vannevar Bush Faculty Fellowship (award N00014-22-1-2790).

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

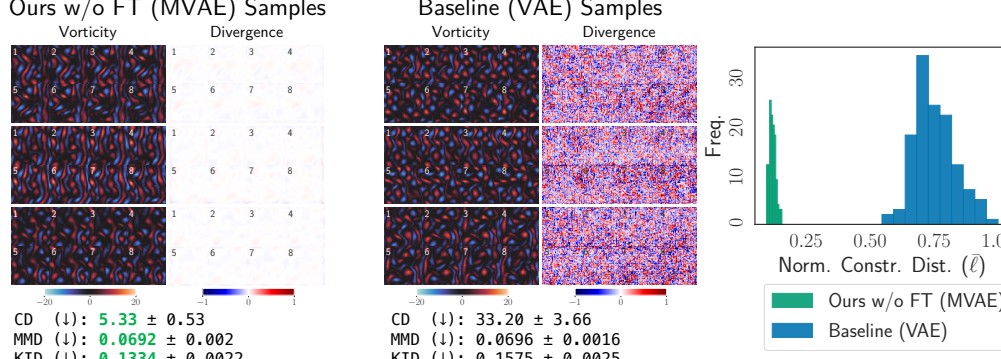

Figure 8: Mirror VAE (MVAE) vs. VAE, demonstrated on the divergence-free constraint. Baseline samples are obtained from a VAE trained in the original space, while MVAE samples are obtained by sampling from a VAE trained in the mirror space and then mapping those samples back to the original space via the learned inverse mirror map. Three samples are shown (vorticity on the left and divergence on the right) for each method. (Recall that each image consists of eight state snapshots; here we have labeled the number of each snapshot.) The vorticity fields show that the visual statistics of both generated distributions are extremely similar, but the corresponding divergence fields are drastically different. The MVAE samples are much closer to satisfying 0-divergence everywhere. As further evidence, the histograms show that normalized constraint distances of MVAE samples are significantly lower. We also report the mean $\pm$ std. dev. constraint distance (CD), computed as $100\lambda_{\mathrm{constr}}\ell$, as well as the MMD and KID. All three metrics were estimated with 10000 generated and true samples. The MVAE leads to improved constraint satisfaction and distribution-matching accuracy compared to a vanilla VAE. This experiment demonstrates how a NAMM can be used to constrain generative models besides diffusion models.

## A    NAMMs FOR CONSTRAINED VAEs

Our approach is compatible with any generative model, not just diffusion models. Once a NAMM is trained, any generative model can be trained in the learned mirror space and its samples mapped back to the constrained space via the learned inverse mirror map. In this appendix, we apply our approach to training a variational autoencoder (VAE) that satisfies the divergence-free constraint, comparing a VAE trained in the learned mirror space ("MVAE") to a VAE trained in the original data space.

### A.1    IMPROVED CONSTRAINT SATISFACTION WITH A MIRROR VAE

For this experiment, we trained a VAE in the mirror space induced by the learned mirror map that was trained for the divergence-free constraint (without finetuning). We call this the "mirror VAE," or MVAE, approach. As a baseline, we trained the same VAE architecture on the original divergence-free data without transformation. We note that the same data are used to train both the MVAE and VAE; the only difference is that the MVAE is trained in the mirror space, while the VAE is trained in the original space. The training procedure was otherwise the same for both the MVAE and VAE.

Figure 8 shows samples from the MVAE and VAE. We ensured that the total training time of the MVAE did not exceed that of the VAE (on the same hardware, $4\times$ A100 GPUs). The VAE was trained for 3500 epochs, and the MVAE was trained for 600 epochs (following 100 epochs of NAMM training). Both approaches produce visually similar samples, yet the images of the divergence field and histograms for this constraint distance show that the MVAE leads to overall better constraint satisfaction. Furthermore, in terms of MMD and KID, the MVAE distribution is even closer to the true data distribution.

### A.2    VAE IMPLEMENTATION DETAILS

We used a convolutional autoencoder architecture consisting of five convolutional layers with GELU activation functions in the encoder and decoder. Our implementation is borrowed from the autoencoder

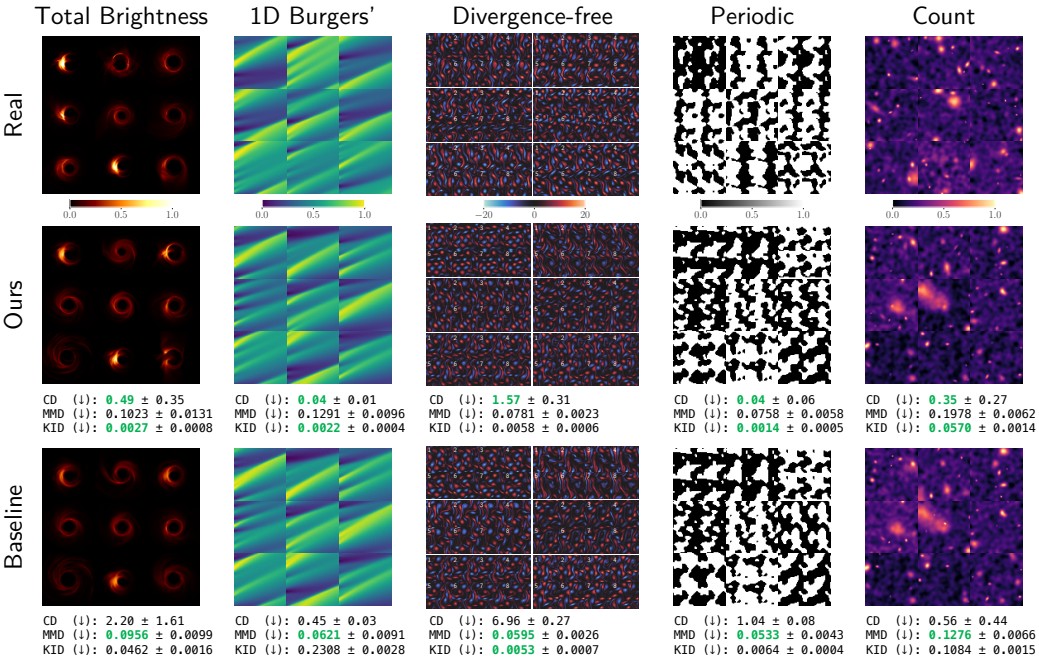

Figure 9: Comparisons of generated distributions. Nine (six for the divergence-free constraint) samples are shown for each distribution. The constraint distance (CD), MMD, and KID values are repeated here from Tab. 1 for ease of comparison. Qualitatively, our approach gives samples that are visually very similar to real samples and baseline samples.

tutorial of Lippe (2024). We set the number of latent dimensions to $128$ and the number of features in the first layer of the encoder to $64$. For training, we followed the $\beta$-VAE training objective (Higgins et al., 2017), which consists of two terms: one to increase the likelihood of training data under the VAE probabilistic model and one to minimize the KL divergence from the latent distribution to a Gaussian prior. The latter is weighted by a scalar $\beta > 0$. For our purposes, the maximum-likelihood term corresponds to a mean-squared-error (MSE) reconstruction loss, and we set $\beta = 0.001$. We used the Adam optimizer with a learning rate of $0.0002$.

## B  SAMPLE COMPARISONS

Figure 9 shows random samples from our approach and the baseline DM approach, corresponding to the results shown in Figure 3. We show more samples in this appendix to give a sense of the visual similarity between the samples generated with our approach and those generated with the baseline approach. We emphasize again that our samples are much closer to the constraint set despite being visually indistinguishable from baseline samples.

## C  CONSTRAINT HYPERPARAMETERS

Here we provide an ablation study of the constraint hyperparameters, $\lambda_{\text{constr}}$ and $\sigma_{\max}$, for all the demonstrated constraints. We performed a hyperparameter sweep across all combinations of $\lambda_{\text{constr}} \in [0.01, 0.1, 1.0]$ and $\sigma_{\max} \in [0.001, 0.1, 0.5]$. For each setting, we trained a NAMM and then an MDM, and then we evaluated the constraint satisfaction and distribution-matching accuracy of the MDM samples. Table 2 reports the constraint distance, MMD, and KID metrics for all hyperparameter settings and constraints.

These results illustrate how performance changes with respect to $\lambda_{\text{constr}}$ and $\sigma_{\max}$. We observe that increasing $\lambda_{\text{constr}}$ generally leads to lower constraint distances for any constraint. For some constraints, there is also an improvement in MMD/KID as $\lambda_{\text{constr}}$ increases (e.g., 1D Burgers'), whereas in other

cases, there seems to be a tradeoff between constraint satisfaction and distribution-matching accuracy (e.g., Count). We observe that the metrics change non-monotonically as a function of $\sigma_{\max}$. This may be attributed to the fact that the training objective has a nonlinear dependence on $\sigma_{\max}$. When applying our method to a new constraint, one can run a similar hyperparameter sweep to choose values that lead to the best combination of constraint satisfaction and distribution-matching accuracy.

| Constraint | $\lambda_{\text{constr}}$ | Metric | $\sigma_{\max} = 0.001$ | $\sigma_{\max} = 0.1$ | $\sigma_{\max} = 0.5$ |
|---|---|---|---|---|---|
| Total Brightness | $\lambda_{\text{constr}} = 0.01$ | CD ($\downarrow$) | $0.66 \pm 0.38$ | $\underline{0.55 \pm 0.45}$ | $0.66 \pm 0.57$ |
| | | MMD ($\downarrow$) | $0.0517 \pm 0.0060$ | $\underline{0.0549 \pm 0.0067}$ | $0.0757 \pm 0.0055$ |
| | | KID ($\downarrow$) | $0.0386 \pm 0.0025$ | $\underline{0.0078 \pm 0.0014}$ | $0.0062 \pm 0.0008$ |
| | $\lambda_{\text{constr}} = 0.1$ | CD ($\downarrow$) | $0.03 \pm 0.02$ | $0.02 \pm 0.01$ | $0.22 \pm 0.03$ |
| | | MMD ($\downarrow$) | $1.1723 \pm 0.0006$ | $1.1723 \pm 0.0006$ | $1.1727 \pm 0.0006$ |
| | | KID ($\downarrow$) | $0.4372 \pm 0.0043$ | $0.5275 \pm 0.0045$ | $0.6396 \pm 0.0047$ |
| | $\lambda_{\text{constr}} = 1.0$ | CD ($\downarrow$) | $0.07 \pm 0.02$ | $0.03 \pm 0.01$ | $0.11 \pm 0.01$ |
| | | MMD ($\downarrow$) | $1.1721 \pm 0.0006$ | $1.1722 \pm 0.0006$ | $1.1727 \pm 0.0006$ |
| | | KID ($\downarrow$) | $0.5052 \pm 0.0044$ | $0.5191 \pm 0.0045$ | $0.6151 \pm 0.0046$ |
| 1D Burgers' | $\lambda_{\text{constr}} = 0.01$ | CD ($\downarrow$) | $0.20 \pm 0.03$ | $0.27 \pm 0.03$ | $0.26 \pm 0.03$ |
| | | MMD ($\downarrow$) | $0.0488 \pm 0.0090$ | $0.0631 \pm 0.0095$ | $0.0945 \pm 0.0078$ |
| | | KID ($\downarrow$) | $0.0483 \pm 0.0012$ | $0.1111 \pm 0.0018$ | $0.1142 \pm 0.0020$ |
| | $\lambda_{\text{constr}} = 0.1$ | CD ($\downarrow$) | $0.20 \pm 0.03$ | $0.20 \pm 0.02$ | $0.18 \pm 0.02$ |
| | | MMD ($\downarrow$) | $0.0784 \pm 0.0075$ | $0.0682 \pm 0.0071$ | $0.1069 \pm 0.0062$ |
| | | KID ($\downarrow$) | $0.0434 \pm 0.0014$ | $0.0495 \pm 0.0016$ | $0.0499 \pm 0.0015$ |
| | $\lambda_{\text{constr}} = 1.0$ | CD ($\downarrow$) | $0.20 \pm 0.03$ | $\underline{0.07 \pm 0.01}$ | $0.17 \pm 0.02$ |
| | | MMD ($\downarrow$) | $0.1159 \pm 0.0078$ | $\underline{0.1092 \pm 0.0090}$ | $0.2169 \pm 0.0084$ |
| | | KID ($\downarrow$) | $0.0494 \pm 0.0012$ | $\underline{0.0026 \pm 0.0004}$ | $0.0144 \pm 0.0008$ |
| Divergence-free | $\lambda_{\text{constr}} = 0.01$ | CD ($\downarrow$) | $4.17 \pm 0.29$ | $5.17 \pm 0.33$ | $6.01 \pm 0.35$ |
| | | MMD ($\downarrow$) | $0.0493 \pm 0.0029$ | $0.0484 \pm 0.0025$ | $0.0522 \pm 0.0018$ |
| | | KID ($\downarrow$) | $0.0007 \pm 0.0002$ | $0.0029 \pm 0.0004$ | $0.0062 \pm 0.0008$ |
| | $\lambda_{\text{constr}} = 0.1$ | CD ($\downarrow$) | $3.37 \pm 0.30$ | $3.16 \pm 0.33$ | $3.56 \pm 0.31$ |
| | | MMD ($\downarrow$) | $0.0593 \pm 0.0026$ | $0.0569 \pm 0.0031$ | $0.0588 \pm 0.0030$ |
| | | KID ($\downarrow$) | $0.0029 \pm 0.0006$ | $0.0009 \pm 0.0002$ | $0.0020 \pm 0.0003$ |
| | $\lambda_{\text{constr}} = 1.0$ | CD ($\downarrow$) | $2.42 \pm 0.36$ | $2.28 \pm 0.18$ | $\underline{2.61 \pm 0.21}$ |
| | | MMD ($\downarrow$) | $0.0554 \pm 0.0037$ | $0.0548 \pm 0.0031$ | $\underline{0.0593 \pm 0.0019}$ |
| | | KID ($\downarrow$) | $0.0031 \pm 0.0005$ | $0.0031 \pm 0.0004$ | $\underline{0.0029 \pm 0.0004}$ |
| Periodic | $\lambda_{\text{constr}} = 0.01$ | CD ($\downarrow$) | $0.25 \pm 0.40$ | $0.28 \pm 0.12$ | $0.30 \pm 0.15$ |
| | | MMD ($\downarrow$) | $0.2133 \pm 0.0057$ | $0.0484 \pm 0.0041$ | $0.0507 \pm 0.0039$ |
| | | KID ($\downarrow$) | $0.0215 \pm 0.0018$ | $0.0006 \pm 0.0002$ | $0.0019 \pm 0.0004$ |
| | $\lambda_{\text{constr}} = 0.1$ | CD ($\downarrow$) | $0.35 \pm 0.29$ | $0.15 \pm 0.09$ | $0.35 \pm 0.20$ |
| | | MMD ($\downarrow$) | $0.0795 \pm 0.0059$ | $0.0621 \pm 0.0045$ | $0.1013 \pm 0.0060$ |
| | | KID ($\downarrow$) | $0.0054 \pm 0.0008$ | $0.0015 \pm 0.0005$ | $0.0044 \pm 0.0008$ |
| | $\lambda_{\text{constr}} = 1.0$ | CD ($\downarrow$) | $0.30 \pm 0.11$ | $0.05 \pm 0.06$ | $0.00 \pm 0.00$ |
| | | MMD ($\downarrow$) | $0.1092 \pm 0.0056$ | $\underline{0.0856 \pm 0.0057}$ | $0.7822 \pm 0.0037$ |
| | | KID ($\downarrow$) | $0.0049 \pm 0.0007$ | $\underline{0.0044 \pm 0.0009}$ | $0.8408 \pm 0.0020$ |
| Count | $\lambda_{\text{constr}} = 0.01$ | CD ($\downarrow$) | $0.73 \pm 0.55$ | $\underline{0.65 \pm 0.49}$ | $0.61 \pm 0.47$ |
| | | MMD ($\downarrow$) | $0.0715 \pm 0.0057$ | $\underline{0.0449 \pm 0.0030}$ | $0.2119 \pm 0.0067$ |
| | | KID ($\downarrow$) | $0.0084 \pm 0.0006$ | $\underline{0.0514 \pm 0.0011}$ | $0.0266 \pm 0.0011$ |
| | $\lambda_{\text{constr}} = 0.1$ | CD ($\downarrow$) | $0.13 \pm 0.10$ | $0.59 \pm 0.45$ | $0.52 \pm 0.41$ |
| | | MMD ($\downarrow$) | $0.1773 \pm 0.0023$ | $0.0884 \pm 0.0058$ | $0.0520 \pm 0.0032$ |
| | | KID ($\downarrow$) | $0.5006 \pm 0.0040$ | $0.0594 \pm 0.0012$ | $0.0086 \pm 0.0004$ |
| | $\lambda_{\text{constr}} = 1.0$ | CD ($\downarrow$) | $0.01 \pm 0.01$ | $0.02 \pm 0.01$ | $0.03 \pm 0.01$ |
| | | MMD ($\downarrow$) | $0.6346 \pm 0.0022$ | $0.6774 \pm 0.0024$ | $0.7988 \pm 0.0029$ |
| | | KID ($\downarrow$) | $0.4995 \pm 0.0018$ | $0.4957 \pm 0.0020$ | $0.4786 \pm 0.0016$ |

Table 2: Ablation study of constraint loss hyperparameters. Constr. dist. (CD) $= 100\lambda_{\text{constr}}\ell$. MMD and KID are metrics for distribution-matching accuracy. The mean $\pm$ std. dev. of each metric was estimated with 10000 samples. For each constraint, the NAMM and MDM were trained for as many epochs (before finetuning) as reported in Table 4. The hyperparameter setting used for the main paper results reported in Table 1 is underlined for each constraint (the results here are slightly different due to randomness in the training runs). These results, which were attained without finetuning, give a sense of how performance changes with respect to $\lambda_{\text{cosntr}}$ and $\sigma_{\max}$ for each constraint.

# D EXPERIMENT DETAILS

## D.1 IMPLEMENTATION

**MDM score model** For training the score model $s_\theta$ in the learned mirror space, we followed the implementation of Song et al. (2021a). We used the NCSN++ architecture with 64 filters in the first layer and the VP SDE with $\beta_{\min} = 0.1$ and $\beta_{\max} = 20$. Training was done using the Adam optimizer with a learning rate of 0.0002 and gradient norm clipping with a threshold of 1.

**NAMM** For $\mathbf{g}_\phi$, we followed the implementation of the gradient of a strongly-convex ICNN of Tan et al. (2023), configuring the ICNN to be 0.9-strongly convex. Following the settings of CycleGAN (Zhu et al., 2017), $\mathbf{f}_\psi$ was implemented as a ResNet-based generator with 6 residual blocks and 32 filters in the last convolutional layer. For all constraints except the divergence-free constraint, we had the ResNet-based generator output the residual image (i.e., $\mathbf{f}_\psi(\tilde{\mathbf{x}}) = \text{ResNet}(\tilde{\mathbf{x}}) + \tilde{\mathbf{x}}$). We found that for the divergence-free constraint, a non-residual-based inverse map (i.e., $\mathbf{f}_\psi(\tilde{\mathbf{x}}) = \text{ResNet}(\tilde{\mathbf{x}})$) achieves better constraint loss. The NAMM was trained using Adam optimizer with a learning rate of 0.001 for the divergence-free constraint and a learning rate of 0.0002 for all other constraints.

Table 3 shows the hyperparameter choices for each constraint. The regularization weight $\lambda_{\text{reg}}$ in the NAMM objective (Equation 1) was fixed at 0.001. We used 3 ICNN layers for images $64 \times 64$ or smaller and 2 ICNN layers for images $128 \times 128$ or larger for the sake of efficiency. These hyperparameter values do not need to be heavily tuned, as we chose these settings through a coarse parameter search (e.g., trying $\lambda_{\text{constr}} = 0.01$ or $\lambda = 1$ to see which would lead to reasonable loss curves).

|  | Num. ICNN layers | $\sigma_{\max}$ | $\lambda_{\text{constr}}$ |
| --- | --- | --- | --- |
| Total Brightness | 3 | 0.1 | 0.01 |
| 1D Burgers' | 3 | 0.1 | 1 |
| Divergence-free | 2 | 0.5 | 1 |
| Periodic | 3 | 0.1 | 1 |
| Count | 2 | 0.1 | 0.01 |

Table 3: NAMM hyperparameter values for each constraint in our experiments.

The main results shown in Figure 3 were taken from the finetuned NAMM, ensuring that the total training time of the NAMM, MDM, and finetuning did not exceed the total training time of the baseline DM. While we kept track of the validation loss, this was not used to determine stopping time. We found that the NAMM training and MDM training were not prone to overfitting, so we chose the total number of epochs based on observing a reasonable level of convergence of the loss curves. We found that some overfitting is possible during finetuning but did not perform early stopping. All results were obtained from unseen test data because we fed random samples from the MDM into the inverse map and made sure not to use the same random seed as the one used to generate finetuning data. For all constraints, we generated 12800 training examples from the MDM for finetuning.

Table 4 details the exact number of training epochs for each constraint. Figure 4 in the main results compares the constraint distances of our method without finetuning, our method with finetuning, and the baseline DM as a function of compute time.

**Mirror map parameterization ablation study** For the comparison of parameterizing the mirror map as the gradient of an ICNN versus as a ResNet-based generator (Figure 7), we used a ResNet-based generator that outputs the residual image. This means that the inverse mirror map was parameterized as a residual-based network ($\mathbf{f}_\psi(\tilde{\mathbf{x}}) = \text{ResNet}(\tilde{\mathbf{x}}; \psi) + \tilde{\mathbf{x}}$), and so was the ResNet-based forward mirror map ($\mathbf{g}_\phi(\mathbf{x}) = \text{ResNet}(\mathbf{x}; \phi) + \mathbf{x}$).

## D.2 ABLATION OF SEQUENCE OF NOISY MIRROR DISTRIBUTIONS

Recall that the NAMM training objective involves optimizing $\mathbf{f}_\psi$ over the sequence of noisy mirror distributions defined in Equation 2. Thus instead of considering a single noisy mirror distribution

| | NAMM epochs (before FT) | MDM epochs | FT epochs | DM epochs |
|---|---|---|---|---|
| Total Brightness | 30 | 200 | 1000 | 450 |
| 1D Burgers' | 100 | 300 | 700 | 1500 |
| Divergence-free | 100 | 700 | 500 | 2000 |
| Periodic | 50 | 300 | 1000 | 1000 |
| Count | 50 | 300 | 700 | 1500 |

Table 4: Number of training epochs of the NAMM, MDM, finetuning, and DM used for the results in Fig. 3. These were chosen so that our method (including the NAMM training, MDM training, finetuning data generation, and finetuning) did not take longer to train than the DM.

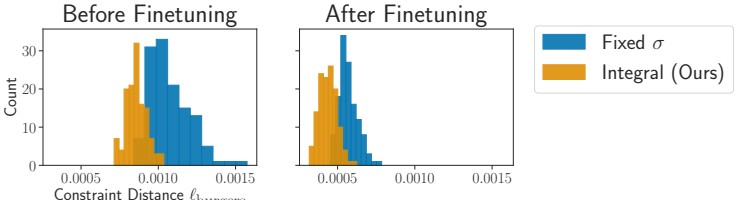

Figure 10: Fixed $\sigma$ vs. integrating over $[0, \sigma_{\max}]$. The NAMM objective involves optimizing $\mathbf{f}_\psi$ over the sequence of noisy mirror distributions defined in Eq. 2. We compare this approach of integrating over $\sigma \in [0, \sigma_{\max}]$ to setting a fixed noise standard deviation $\sigma = \sigma_{\max}$ in the context of the 1D Burgers' constraint (here $\sigma_{\max} = 0.1$). Both before and after finetuning, the constraint distances $\ell_{\text{burgers}}$ of inverted MDM samples are smaller if the NAMM was trained with varying $\sigma$. The histograms show the constraint distances of 128 samples from each method.

$(\mathbf{g}_\phi)_\sharp p_{\text{data}} * \mathcal{N}(\mathbf{0}, \sigma^2 \mathbf{I})$ with a fixed noise level $\sigma$, we perturb samples from $(\mathbf{g}_\phi)_\sharp p_{\text{data}}$ with varying levels of Gaussian noise with variances ranging from 0 to $\sigma_{\max}^2$. In Figure 10, we compare this choice, which involves integrating over $\sigma \in [0, \sigma_{\max}]$, to the use of a fixed $\sigma = \sigma_{\max}$.

### D.3 DATA ASSIMILATION WITH MIRROR DPS

Following the original DPS (Chung et al., 2022), we use a hyperparameter $\zeta \in \mathbb{R}_{>0}$ to re-weight the time-dependent measurement likelihood. At diffusion time $t$, the measurement weight is given by

$$\zeta(t) := \zeta / \|\mathbf{y} - \mathcal{A}(\hat{\mathbf{x}}_0)\|_\Gamma , \qquad (8)$$

where $\hat{\mathbf{x}}_0 := \mathbf{f}_\psi \left( \mathbb{E}[\tilde{\mathbf{x}}(0) \mid \tilde{\mathbf{x}}(t)] \right)$. Here we assume that the measurement process has the form

$$\mathbf{y} = \mathcal{A}(\mathbf{x}^*) + \mathbf{n}, \quad \mathbf{n} \sim \mathcal{N}(\mathbf{0}_m, \Gamma) \qquad (9)$$

for some unknown source image $\mathbf{x}^* \in \mathbb{R}^d$, where $\mathcal{A} : \mathbb{R}^d \to \mathbb{R}^m$ is a known forward operator, and $m \times m$ is the known noise covariance matrix $\Gamma$. Higher values of $\zeta$ impose greater measurement consistency, but setting $\zeta$ too high can cause instabilities and artifacts. The data assimilation results in Figure 5 used $\zeta = 0.1$ and constraint-guidance weight equal to 200. Figure 11 shows results for the same tasks but different values of $\zeta$ in DPS and the constraint-guidance weight in CG-DPS.

### E MEASURES OF DISTANCE BETWEEN DISTRIBUTIONS

#### E.1 MMD

The maximum mean discrepancy (MMD) between two distributions is computed by embedding both distributions into a reproducing kernel Hilbert space (RKHS) and using samples to estimate the resulting distance. We use the popular Gaussian radial basis function (RBF) kernel to construct the RKHS, setting the length scale $\sigma$ as

$$\sqrt{\text{median} \left( \left\{ \|\mathbf{x}^{(i)} - \mathbf{x}^{(j)}\|_2^2 \right\} \right) / 2}, \qquad (10)$$

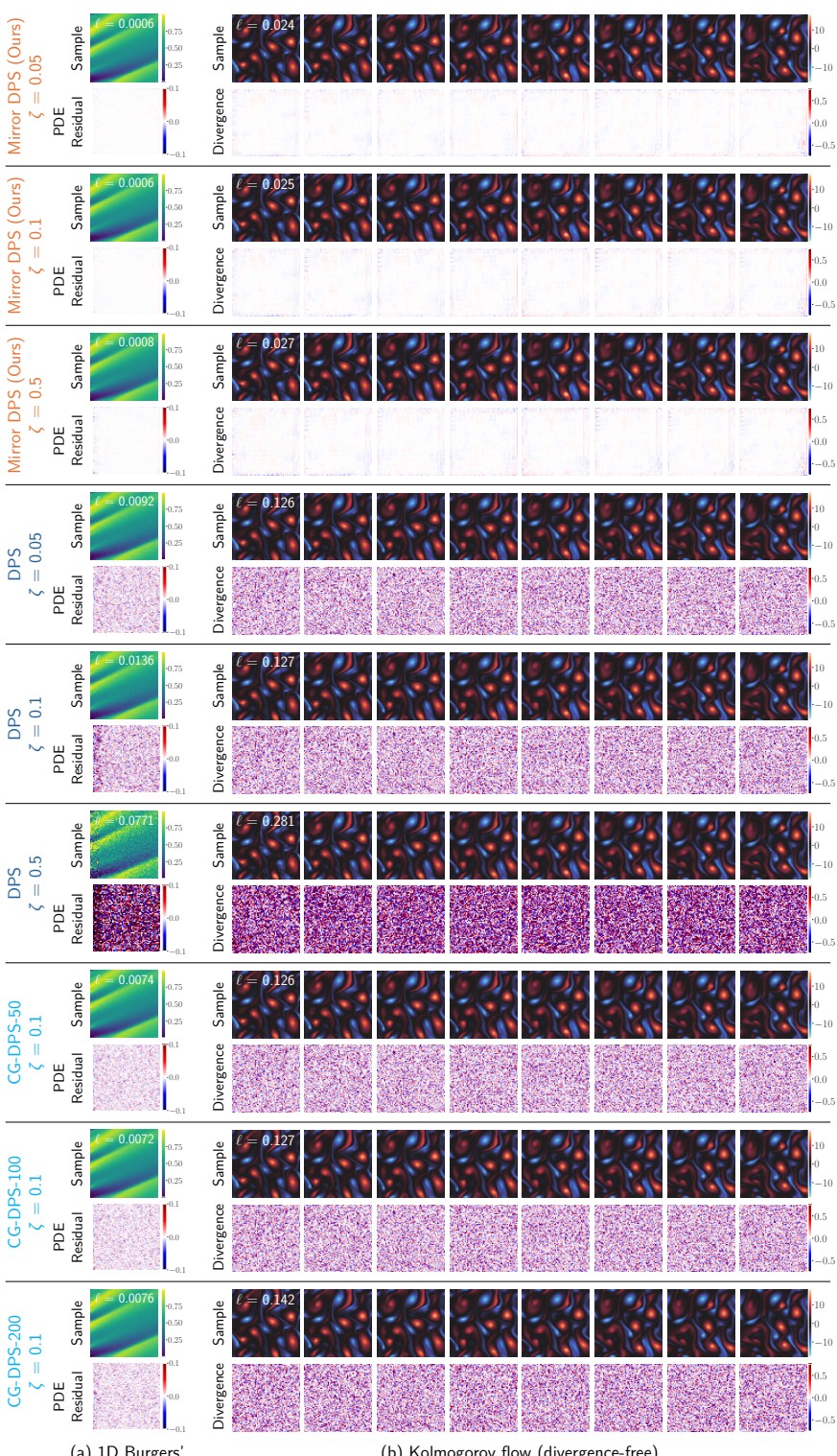

(a) 1D Burgers'  (b) Kolmogorov flow (divergence-free)

Figure 11: Data assimilation results for different values of $\zeta$ and constraint-guidance strengths. E.g., "CG-DPS-50" refers to CG-DPS with a constraint-guidance weight of $50$. These highlight that even for different values of $\zeta$, the constraint errors (i.e., PDE residual or absolute divergence) of our solutions are much smaller than those of the baseline solutions. Furthermore, changing the constraint-guidance weight of CG-DPS does not meaningfully change the level of constraint satisfaction.

i.e., the square root of half the median of the pairwise squared Euclidean distances in the dataset $\left\{\mathbf{x}^{(i)}\right\}_{i=1}^{n}$. This is a popular choice in previous work (Briol et al., 2019) and has been theoretically and empirically justified Garreau et al. (2017).

Our MMD implementation is based on the code provided for the work of Sutherland et al. (2016) (Sutherland, 2018). We estimated mean and standard deviation empirically with 50 random subsets of 1000 samples from each dataset. The length scale was estimated (Equation 10) for each subset using the samples in the true subset. In total, the generated and true datasets contained 10000 samples each. Held-out test images were used as true samples for all constraints, except for total brightness (due to a lack of test images in the dataset, training images were used for this constraint only).

### E.2   KID

The Kernel Inception Distance (KID) between two distributions is based on the Inception v3 features evaluated for samples from both distributions. Following standard practice, we use the 2048-dimensional final average pooling features. The KID is computed as the squared MMD (using a polynomial kernel) between the two embedded distributions. It has several advantages over the Fréchet Inception Distance (FID) (Heusel et al., 2017), including being unbiased and more sample-efficient (Bińkowski et al., 2018). We note, however, that the Inception network was trained on natural images, so both KID and FID are not perfect metrics for the types of data we consider in this work, such as physics-based simulation outputs.

KID evaluation is based on the `gan-metrics-pytorch` repository (Fatir, 2021), using the same Inception v3 weights as those used in the official TensorFlow implementation of FID (Heusel et al., 2017). We evaluated KID with the same samples that were used for MMD. Since the Inception network takes RGB images as input, we represented the samples as grayscale images converted to RGB. For all the constraint datasets except for the divergence-free Kolmogorov flows, we clipped the image pixel values to $[0, 1]$ before converting them to RGB. For the divergence-free data, we clipped the values in the vorticity images to $[-20, 20]$ and then rescaled this range to $[0, 1]$.

### E.3   FID

Although FID is a biased finite-sample estimator and heavily depends on the number of samples, it is a popular metric for evaluating generative models. Table 5 reports FID values in addition to the MMD and KID values in Table 1. We find that the rankings of our method before finetuning, our method after finetuning, and the baseline method are consistent with the rankings when using KID in Table 1. Again we note that FID and KID are based on Inception features that were tuned to natural images, so they are not the most reliable measures of distance between distributions for these particular image datasets.

|             | Total Brightness | 1D Burgers' | Divergence-free | Periodic | Count     |
|-------------|------------------|-------------|-----------------|----------|-----------|
| Ours w/o FT | 5.02             | 3.36        | **2.21**        | 2.26     | **24.18** |
| Ours        | **4.58**         | **2.04**    | 3.97            | **1.99** | 37.65     |
| Baseline    | 42.25            | 140.70      | 3.41            | 6.42     | 69.76     |

Table 5: FID values ($\downarrow$) associated with the comparisons in Tab. 1. According to FID, our approach consistently outperforms the baseline DM approach in matching the true distribution. Finetuning even sometimes improves FID while improving constraint satisfaction (see Tab. 1 for constraint distances).

Our FID implementation is borrowed from the `pytorch-fid` codebase (Seitzer, 2020). Per standard practice, we estimated FID with 50000 generated samples and 50000 true samples. As was the case with MMD and KID, held-out test images were used as true samples, except that training images were used for the total brightness data. We pre-processed the samples for the Inception network in the same way we did for KID.

## F    CONSTRAINT DETAILS

Here we provide details about each demonstrated constraint and its corresponding dataset.

**Total brightness**    The total brightness, or total flux, of a discrete image $\mathbf{x} \in \mathbb{R}^d$ is simply the sum of its pixel values: $V(\mathbf{x}) := \sum_{i=1}^{d} \mathbf{x}_i$. We use the constraint distance function

$$\ell_{\text{flux}}(\mathbf{x}) := \left| V(\mathbf{x}) - \bar{V} \right|,$$

where $\bar{V} \in \mathbb{R}_{\geq 0}$ is the target total brightness. The dataset used for this constraint contains images from general relativistic magneto-hydrodynamic (GRMHD) simulations (Wong et al., 2022) of Sgr A* with a fixed field of view. The images (originally $400 \times 400$) were resized to $64 \times 64$ pixels and rescaled to have a total flux of 120. The dataset consists of 100000 training images and 100 validation images.

**1D Burgers'**    Burgers' equation (Bateman, 1915; Burgers, 1948) is a nonlinear PDE that is a useful model for fluid mechanics. We consider the equation for a viscous fluid $u = u(t, x)$ in one-dimensional space:

$$\frac{\partial u}{\partial t} + u \frac{\partial u}{\partial x} = \nu \frac{\partial^2 u}{\partial x^2}, \tag{11}$$

where $u(0, x)$ is some initial condition $u_0(x)$, and $\nu \in \mathbb{R}_{\geq 0}$ is the viscosity coefficient. We use Crank-Nicolson (Crank & Nicolson, 1947) to discretize and approximately solve Equation 11 by representing the solution on an $n_x \times n_t$ grid, where $n_x$ is the spatial discretization, and $n_t$ is the number of snapshots in time. Given an $n_x \times n_t$ image, we wish to verify that it could be a solution to Equation 11 with the Crank-Nicolson discretization. Letting $\mathbf{x} \in \mathbb{R}^{n_x \times n_t}$ denote the 2D image, we formulate the following distance function for evaluating agreement with the Crank-Nicolson solver:

$$\ell_{\text{burgers}}(\mathbf{x}) := \frac{1}{n_t - 1} \sum_{t=0}^{n_t - 2} \left\| \mathbf{x}[:, t+1] - f_{\text{C-N}}(\mathbf{x}[:, t]) \right\|_1,$$

where $f_{\text{C-N}} : \mathbb{R}^{n_x} \to \mathbb{R}^{n_x}$ outputs the snapshot at the next time using Crank-Nicolson, and Pythonic notation is used for simplicity. Note that a finite-differences loss as proposed for physics-informed neural networks (PINNs) Raissi et al. (2019) would also work, but then our data would have non-negligible constraint distances since Crank-Nicolson solutions do not strictly follow a low-order finite-differences approximation.

Using a Crank-Nicolson solver (Crank & Nicolson, 1947) implemented with Diffrax (Kidger, 2022), we numerically solved the 1D Burgers' equation (Equation 11) with viscosity coefficient $\nu = 0.5$. The initial conditions were sampled from a Gaussian process based on a Matérn kernel with smoothness parameter 1.5 and length scale equal to 1.0. We discretized the spatiotemporal domain into a $64 \times 64$ grid covering the spatial extent $x \in [0, 10]$ and time interval $t \in [0, 8]$. We ran Crank-Nicolson with a time step of $\Delta t = 0.025$ and saved every fifth step for a total of 64 snapshots. We followed this process to create our 1D Burgers' dataset of 10000 training images and 1000 validation images.

**Divergence-free**    The study of fluid dynamics often involves incompressible, or divergence-free, fluids. Letting $\mathbf{u} = \mathbf{u}(x, y, t)$ be the time-dependent trajectory of a 2D velocity field, the divergence-free constraint says that $\nabla \cdot \mathbf{u} = \frac{\partial \mathbf{u}_x}{\partial x} + \frac{\partial \mathbf{u}_y}{\partial y} = 0$. We assume an $n_x \times n_y$ spatial grid and represent trajectories as two-channel (for the two velocity components) images showing each $n_x \times n_y$ snapshot for a total of $n_t$ snapshots. Such an image $\mathbf{x}$ has a corresponding image of the divergence field $\text{div}(\mathbf{x})$, which has the same size as $\mathbf{x}$ and represents the divergence of the trajectory. We formulate the following distance function that penalizes non-zero divergence:

$$\ell_{\text{div}}(\mathbf{x}) := \left\| \text{div}(\mathbf{x}) \right\|_1.$$

We created a dataset of Kolmogorov flows, which satisfy a Navier-Stokes PDE, to demonstrate the divergence-free constraint. The Navier-Stokes PDEs are ubiquitous in fields including fluid dynamics, mathematics, and climate modeling and have the following form:

$$\frac{\partial \mathbf{u}}{\partial t} = -\mathbf{u}\nabla\mathbf{u} + \frac{1}{Re}\nabla^2\mathbf{u} - \frac{1}{\rho}\nabla p + \mathbf{f}\nabla \cdot \mathbf{u} = 0,$$

where $\mathbf{u} = \mathbf{u}(x, y, t)$ is the 2D velocity field at spatial location $(x, y)$ and time $t$, $Re$ is the Reynolds number, $\rho$ is the density, $p$ is the pressure field, and $\mathbf{f}$ is the external forcing. Following Kochkov et al. (2021) and Rozet & Louppe (2024), we set $Re = 10^3$, $\rho = 1$, and $\mathbf{f}$ corresponding to Kolmogorov forcing (Chandler & Kerswell, 2013; Boffetta & Ecke, 2012) with linear damping. We consider the spatial domain $[0, 2\pi]^2$ with periodic boundary conditions and discretize it into a $64 \times 64$ uniform grid. We used `jax-cfd` (Kochkov et al., 2021) to randomly sample divergence-free, spectrally filtered initial conditions and then solve the Navier-Stokes equations with the forward Euler integration method with $\Delta t = 0.01$ time units. We saved a snapshot every 20 time units for a total of 8 snapshots in the time interval $[3, 4.6]$. We represent the solution as a two-channel $128 \times 256$ image showing the snapshots in left-to-right order. In total, the dataset consists of 10000 training images and 1000 validation images.

**Periodic** Assuming the constraint that every image is a periodic tiling of $n_{\text{tiles}}$ unit cells, we formulate the following constraint distance for a given image $\mathbf{x}$:

$$\ell_{\text{periodic}}(\mathbf{x}) := \sum_{i=1}^{n_{\text{tiles}}} \frac{1}{n_{\text{tiles}}} \sum_{j=1}^{n_{\text{tiles}}} \|\mathbf{t}_i(\mathbf{x}) - \mathbf{t}_j(\mathbf{x})\|_1 ,$$

which compares each pair of tiles, where $\mathbf{t}_i(\mathbf{x})$ denotes the $i$-th tile in the image. For our experiments, we consider $32 \times 32$ unit cells that are tiled in a $2 \times 2$ pattern to create $64 \times 64$ images. Using the unit-cell generation code of Ogren et al. (2024), we created a dataset of 30000 training images and 300 validation images.

**Count** For the count constraint, we rely on a CNN to estimate the count of a particular object. Letting $f_{\text{CNN}} : \mathbb{R}^d \to \mathbb{R}$ be the trained counting CNN, we turn to the following constraint distance function for a target count $\bar{c}$:

$$\ell_{\text{count}}(\mathbf{x}) := |f_{\text{CNN}}(\mathbf{x}) - \bar{c}| .$$

We demonstrate this constraint with astronomical images that contain a certain number of galaxies. In particular, we simulated $128 \times 128$ images of radio galaxies with background noise Connor et al. (2022), each of which has exactly eight ($\bar{c} = 8$) galaxies with an SNR $\geq 15$ dB. The dataset consists of 10000 training images and 1000 validation images.

To train the counting CNN, we created a mixed dataset with images of $6, 7, 8, 9$, or $10$ galaxies that includes 10000 training images and 1000 validation images for each of the five labels. The CNN architecture was adapted from a simple MNIST classifier (8bitmp3, 2023) with two convolutional layers followed by two dense layers with ReLU activations. The CNN was trained to minimize the mean squared error between the real-valued estimated count and the ground-truth count.

