# OpenReview forum: "Neural Approximate Mirror Maps for Constrained Diffusion Models"
_ICLR.cc/2025/Conference — ICLR 2025 Poster_

### Official Review · Reviewer_FTXm · 2024-10-28

**Soundness:** 3
**Presentation:** 3
**Contribution:** 3
**Rating:** 8
**Confidence:** 3

**Summary:**

This paper proposes a technique to learn a mirror map for general constrained diffusion generation. The resulting neural approximate mirror map transforms the constrained problem domain to an unconstrained space, where the diffusion model is trained. The training loss encourages that the inverse of the mirror map lies within the constraint set. Numerical experiments on various problems demonstrate the efficacy of the proposed technique in enforcing the constraints.

**Strengths:**

- The method is well motivated. Proper constraint satisfaction is challenging in diffusion generation, limiting many important applications in engineering, physics and computer vision.

- The proposed approach is sensible, and to the best of my knowledge novel. It improves the flexibility of mirror diffusion models by obviating the need for analytical mirror maps.

- The experimental results are promising on a wide array of problems.

- The paper overall is well-written and easy to follow.

**Weaknesses:**

- It is unclear how complex of a constraint the proposed method can handle, as the gap between NAMM and the baseline is less significant for the semantic problem.
- It is unclear if there is a systematic way to tune the introduced hyperparameters, and how sensitive the performance is in higher dimensions to $\sigma_{max}$.

**Questions:**

- It appears in Fig. 6 that constraint distance is fairly sensitive to $\sigma_{max}$ and $\lambda_{constr}$. Can authors propose systematic ways to tune the hyperparameters? How does the sensitivity depend on the dimensionality of the problem or the complexity of the constraint?
- Minor comment: the citations in Section 2.1 take up a large portion of the paragraph somewhat reducing readability (top of page 3).
- Minor comment 2: Fig. 2 is never directly referenced in the text.

---

> ### Author Response · Authors · 2024-11-14
>
> Thank you for the positive feedback.
>
> A hyperparameter sweep for $\sigma_\max$ and $\lambda_\text{constr}$ can be used to determine the values that would provide the best performance in terms of constraint distance and distribution-matching accuracy. As detailed in Appendix C.1, we did not extensively search the hyperparameter space and simply looked at $\sigma_\max=0.1, 0.5$ and $\lambda_\text{constr}=0.01, 1$.
>
> Intuitively, it would make sense that $\sigma_\max$ should be increased with a higher dimensionality: because data points are more separated in higher dimensional spaces, adding more noise would ensure the maps are trained accurately at regions between the data points. We did not find a great deal of hyperparameter sensitivity with the dimensionalities considered in our experiments. We believe that a related theoretical direction to pursue is to understand how the best possible constraint satisfaction depends on the complexity of the constraint and how that might inform the choices of $\lambda_\text{constr}$ and $\sigma_\max$.
>
> We believe that the performance gap for the semantic constraint has to do with the gradient-based nature of our approach. One avenue we are exploring is the use of gradient-free methods to optimize the NAMM, which could better handle constraints with irregular or undefined derivatives.
>
> Thank you for the suggestions for improving the text; we will revise the text accordingly.

---

> > ### Author Response · Authors · 2024-11-23
> >
> > Dear Reviewer FTXm,
> >
> > As the end of the discussion period approaches, we would greatly appreciate it if you could confirm whether our response has adequately addressed your concerns.
> >
> > We encourage you to take a look at Appendix C of our revision, in which we have added results of a hyperparameter sweep of $\sigma_\text{max}$ and $\lambda_\text{constr}$ for all benchmark constraints. We found that increasing $\lambda_\text{constr}$ leads to consistent improvements in constraint satisfaction; for many constraints, we expect a tradeoff in distribution-matching accuracy as $\lambda_\text{constr}$ becomes too high. This tradeoff should guide the choice of $\lambda_\text{constr}$. We found that the metrics vary less monotonically with changes in $\sigma_\text{max}$. When applying our method to a new constraint, we recommend doing a similar hyperparameter sweep to optimize the desired metrics.
> >
> > Please let us know if you have any remaining questions or concerns. If your concerns have been resolved, we kindly ask you to consider raising your rating. Thank you again for your time and efforts in reviewing our manuscript.
> >
> > Sincerely,
> >
> > The Authors

---

> > > ### Comment · Reviewer_FTXm · 2024-11-26
> > > **Response to authors**
> > >
> > > Thank you for addressing my concerns, I am raising my rating accordingly.

---

> > > > ### Author Response · Authors · 2024-11-26
> > > >
> > > > Dear Reviewer FTXm,
> > > >
> > > > We are glad that your concerns are addressed, and we thank you for raising your rating accordingly. We greatly appreciate your thoughtful efforts in reviewing our manuscript!
> > > >
> > > > Sincerely,
> > > >
> > > > The Authors

---

### Official Review · Reviewer_iSD2 · 2024-11-02

**Soundness:** 3
**Presentation:** 2
**Contribution:** 3
**Rating:** 6
**Confidence:** 3

**Summary:**

This paper presents neural approximate mirror maps (NAMMs) to enforce soft constraints for diffusion models (DMs). NAMMs employ two neural networks to learn a mirror map that maps constrained points into the mirror space, and its inverse that transforms data back to the constraint set. A mirror diffusion model (MDM) can be trained in the learned mirror space, and its generated samples can be mapped to the constraint set via the inverse map. This method is tested on five benchmark problems, ranging from physics-based, geometric to semantic constraints, and the results show the proposed method improve constraint satisfaction compared to a vanilla unconstrained DM. And, this paper also demonstrates NAMMs leads to less constraint violation when solving constrained inverse problems.

**Strengths:**

1. NAMMs generalize the concept of true mirror maps to learn approximate mirror maps to handle non-convex constraints.
2. NAMMs can handle physics-based, geometric and semantic constraints, while existing methods are restricted in the types of constraints they can handle.
3. NAMMs not only help diffusion models, but also help VAEs to improve constraint satisfaction, showing the potential to be compatible for other generative models. And NAMMs are also helpful to diffusion-based inverse-problem solvers for solving constrained inverse problems.

**Weaknesses:**

1. Theoretically, NAMMs lack the guarantee of the existence and uniqueness of the mirror maps when applied to non-convex problems.
2. The proposed method is validated on five benchmark problems in the main experiments to show the superiority of applying NAMMs to diffusion models in generating constrained data. However, in the experiments to solve inverse problems, ablation experiments, and the experiments applied to VAE, this method is only carried out on partial problems and does not fully demonstrate its performance on the three types of constraints mentioned.
3. Finetuning is an important part of the proposed method introduced in section 3. But, it is mentioned in subsection 4.1 that “We show results from a finetuned NAMM, but as shown in Section 4.3, finetuning is often not necessary”. Moreover, in the ablation studies of constraint loss and mirror map parameterization, and experiments about the VAE, fine tuning is not used.
4. The basic unconstrained model is used for comparison, but the comparison with another existing methods dealing with constraints is lacking.

**Questions:**

1.	In this paper, the robustness of the model is enhanced by introducing noise into the mirror space. What does robustness mean here, and are there any experimental results that support the robustness of the method?
2.	In order to fully demonstrate the performance of the method, it is necessary to supplement its experiments on five benchmark problems and an additional baseline model.
3.	If fine-tuning is considered to be one of the important components of the method and one of the contributions of this paper, more experimental support is needed.
4.	On line 1097, there is a clerical error, “a la”.

---

> ### Author Response · Authors · 2024-11-14
>
> Thank you for your feedback and suggestions.
>
> Robustness of our method is enhanced by introducing noise in the mirror space. By robustness, we mean that the inverse mirror map is able to restore a wider region of $\mathbb{R}^d$ to the constraint set. A robust inverse map works not just for points from the data distribution but also for points somewhat off the data manifold. Figure 6 demonstrates the robustness provided by introducing noise in the mirror space. When the noise level $\sigma_\max$ is too low (i.e., $\sigma_\max=0.001$), the constraint distances are higher. This indicates that training the NAMM with too little noise in the mirror space makes it difficult for the inverse mirror map to handle any errors introduced by the mirror diffusion model.
>
> We performed ablation and inverse problem experiments on a subset of the considered constraints for the sake of presentation clarity. We chose to focus on the more complex physics-based constraints to highlight the applicability of our method to physics-constrained inverse problems. We would be happy to include experiments on all five constraints in an appendix.
>
> To our knowledge, there is no other approach that is applicable to all the constraints we consider. Since there was no relevant baseline, we instead considered a modification of the DPS method to act as a baseline we call “constraint-guided DPS.”
>
> We do not consider finetuning to be an essential component of the method. As Figure 4 and Table 1 show, most of the performance is already achieved before finetuning. We suggest finetuning as an optional additional step to further boost the constraint satisfaction. We believe it is a strength of our method that the results are not so dependent on finetuning.

---

> > ### Author Response · Authors · 2024-11-23
> >
> > Dear Reviewer iSD2,
> >
> > As the end of the discussion period approaches, we would greatly appreciate it if you could confirm whether our response has adequately addressed your concerns.
> >
> > In response to your comment about performing ablation studies on all five benchmark constraints, we have added in Appendix C a full ablation study of the constraint hyperparameters for all five benchmark problems. Please refer to the PDF for the full table. Due to time constraints, we trained the MDM for fewer epochs than those used in the main paper results; the table will be updated in the final manuscript. We copy the text of that appendix below for your reference.
> >
> > > Here we provide an ablation study of the constraint hyperparameters, $\lambda_\text{constr}$ and $\sigma_\text{max}$, for all the demonstrated constraints. We performed a hyperparameter sweep across all combinations of $\lambda_\text{constr}\in[0.01, 0.1, 1.0]$ and $\sigma_\text{max}\in[0.001,0.1,0.5]$. For each setting, we trained a NAMM and then an MDM, and then we evaluated the constraint satisfaction and distribution-matching accuracy of the MDM samples. Table 2 reports the constraint distance, MMD, and KID metrics for all hyperparameter settings and constraints.
> >
> > > These results illustrate how performance changes with respect to $\lambda_\text{constr}$ and $\sigma_\text{max}$. We observe that increasing $\lambda_\text{constr}$ generally leads to lower constraint distances for any constraint. For some constraints, there is also an improvement in MMD/KID as $\lambda_\text{constr}$ increases (e.g., 1D Burgers'), whereas in other cases, there seems to be a tradeoff between constraint satisfaction and distribution-matching accuracy (e.g., Count). We observe that the metrics change non-monotonically as a function of $\sigma_\text{max}$. This may be attributed to the fact that the training objective has a nonlinear dependence on $\sigma_\text{max}$. When applying our method to a new constraint, one can run a similar hyperparameter sweep to choose values that lead to the best combination of constraint satisfaction and distribution-matching accuracy.
> >
> > If any questions remain, please let us know, and we will do our best to respond within the remaining time.
> >
> > If your concerns have been resolved, we kindly ask you to consider raising your rating, as this would reflect the improvements made based on your valuable feedback. Thank you again for your time and efforts in reviewing our manuscript.
> >
> > Sincerely,
> >
> > The Authors

---

> > > ### Comment · Reviewer_iSD2 · 2024-11-25
> > > **Response after rebuttal**
> > >
> > > Thank you for your comprehensive response. The consideration for baseline selection is reasonable, and I look forward to the more comprehensive results of NAMM you will provide in the appendix of the revised manuscript. However, I still have questions about fine-tuning. Since it is not an essential part of the method and does not significantly improve the results, why is there a need to introduce fine-tuning in the core method part. Overall, regardless of fine-tuning, NAMM still provides a more universal method, and I will increase my score.

---

> > > > ### Author Response · Authors · 2024-11-26
> > > >
> > > > Dear Reviewer iSD2,
> > > >
> > > > Thank you for considering our response and for incorporating our proposed additional experiments into your review! In response to your point about finetuning, we have added to the manuscript the following language (lines 251-253 in the revised manuscript) in the subsection about finetuning:
> > > >
> > > > > As the ablation study in Section 4.3 shows, finetuning is not an essential component of the method; we suggest it as an optional step for when it is critical to optimize the constraint distance metric.
> > > >
> > > > We sincerely appreciate your valuable feedback.
> > > >
> > > > Sincerely,
> > > >
> > > > The Authors

---

### Official Review · Reviewer_y9Gb · 2024-11-04

**Soundness:** 3
**Presentation:** 3
**Contribution:** 3
**Rating:** 6
**Confidence:** 2

**Summary:**

The paper proposes neural approximate mirror maps for constraint data generation with diffusion models. Compared to typical mirror diffusion models, the mirror map is parameterized by the gradient of ICNN and learned via penalizing the differentiable constraint distance, thereby being applicable to general non-convex constraints. The forward and inverse mirror maps are learned by a combination of cycle consistency loss, constraint loss and regularization loss. Experiments in several settings ranging from physics-based to semantic demonstrate the effectiveness on constraint satisfaction, training efficiency and constrained inverse problem solving.

**Strengths:**

- The method is applicable to more general constraints than previous works.
- The cycle-consistency loss tailored for mirror maps and diffusion models is sound.
- The experiments are conducted on diverse settings, including constrained DPS.
- The ablation studies are comprehensive.

**Weaknesses:**

- The experiments are primarily toy. It is not clear whether the proposed method can scale to high dimensions and apply to domains such as images.

**Questions:**

- The regularization loss is introduced to ensure a unique solution according to the paper. What is the meaning of unique? As the mirror map is parameterized as the gradient of ICNN, the reversibility is already ensured.
- Is the method scalable? For example, can it be applied to the image settings in Reflected Diffusion Models?

---

> ### Author Response · Authors · 2024-11-14
>
> Thank you for the positive feedback and clarifying questions.
>
> We would like to clarify that all of our experiments are conducted on image datasets. For example, the divergence-free constraint is demonstrated on 128x256 images. The NAMM architecture can easily scale to even higher dimensions, and we expect the bottleneck to be the diffusion model itself, which is known to be slow for generating very large images. To overcome this bottleneck, it is possible to train a latent diffusion model in the mirror space, where the latent space would be lower dimensional and thus cheaper to sample in.
>
> Our method can be easily applied to the constraints considered in Reflected Diffusion Models [1], which are convex and have simpler analytical forms in comparison to the physics-based and semantic constraints we consider in the paper. For example, we can easily design a constraint distance for our method to enforce the constraint that pixel values are bounded between two values.
>
> The regularization loss ensures uniqueness of the forward and inverse mirror maps. Just the fact that the forward map is invertible does not ensure uniqueness: as a simple example, the forward map $f(x) = x$ has the inverse $g(x) = x$, but it could be scaled to $f’(x) = 2x$ and have the inverse $g’(x) = 0.5x$. The regularization loss helps resolve this scale/shift degeneracy.
>
> ---
> References:
>
> [1] Aaron Lou and Stefano Ermon. “Reflected Diffusion Models.” ICML 2023.

---

> > ### Author Response · Authors · 2024-11-23
> >
> > Dear Reviewer y9Gb,
> >
> > As the end of the discussion period approaches, we would greatly appreciate it if you could confirm whether our response has adequately addressed your concerns. If you have any remaining questions, please let us know, and we will do our best to respond within the remaining time.
> >
> > If your questions have been addressed, we kindly ask you to consider raising your rating. Thank you again for your time and efforts in reviewing our manuscript.
> >
> > Sincerely,
> >
> > The Authors

---

> > > ### Comment · Reviewer_y9Gb · 2024-11-26
> > >
> > > Thank the authors for the detailed response. My concerns are well addressed. I decide to keep my score of leaning towards acceptance.

---

> > > > ### Author Response · Authors · 2024-11-26
> > > >
> > > > Dear Reviewer y9Gb,
> > > >
> > > > We sincerely thank you for your efforts in reviewing our paper and for your valuable feedback.
> > > >
> > > > Sincerely,
> > > >
> > > > The Authors

---

### Author Response · Authors · 2024-11-14
**Global Response**

We thank the reviewers for their feedback and suggestions. NAMMs offer a way to flexibly incorporate more general constraints than possible with previous methods (y9Gb, iSD2), and we demonstrate this with promising results “on a wide array of problems” (FTXm). All reviewers appreciated the broader impact of our approach, as it addresses the challenge of constrained generative modeling for “important applications in engineering, physics and computer vision” (FTXm). They also recognized the broad applicability of our approach to inverse problems (y9Gb, iSD2) and different generative models (iSD2). Overall, our proposed approach is “sound” (y9Gb) and “novel” (FTXm), and it is backed by a “well-written” paper (FTXm) and “comprehensive” ablation studies (y9Gb). We will address individual reviewers’ questions in the individual responses.

---

### Meta-Review · Area_Chair_oT5Y · 2024-12-18

**Metareview:**

This paper suggests incorporating constraints into diffusion generative models by mapping samples to an unconstrained space, training a generative diffusion process in the unconstrained space, and then sampling by computing the inverse map of samples in the unconstrained space. The mapping into the unconstrained space (mirror map) is defined as gradient of a convex function and is jointly optimized with the inverse (mirror) map to cover the likely sample space in the unconstrained space as well as other desired properties.

The method is applicable to a wider set of constraints compared to previous works (e.g., non-convex sets in R^d), the optimization of the mirror map and its inverse are solid, and there is a rather extensive experimental section. In terms of weaknesses, it is not clear how scalable the method is to higher dimensions (e.g., using gradients of ICNN and the mirror/inverse-mirror computation), and theoretically the mirror and inverse-mirror maps are not guaranteed to be defined uniquely or actually satisfy forward-inverse relation.

**Additional Comments On Reviewer Discussion:**

No additional comments.

---

### Decision · Program_Chairs · 2025-01-22

Accept (Poster)